# MALCOM-PSGD: Inexact Proximal Stochastic Gradient Descent for Communication-Efficient Decentralized Machine Learning

## Abstract

Recent research indicates that frequent model communication stands as a major bottleneck to the efficiency of decentralized machine learning (ML), particularly for large-scale and over-parameterized neural networks (NNs). In this paper, we introduce MALCOM-PSGD, a new decentralized ML algorithm that strategically integrates gradient compression techniques with model sparsification. MALCOM-PSGD leverages proximal stochastic gradient descent to handle the non-smoothness resulting from the $\ell_1$ regularization in model sparsification. Furthermore, we adapt vector source coding and dithering-based quantization for compressed gradient communication of sparsified models. Our analysis shows that decentralized proximal stochastic gradient descent with compressed communication has a convergence rate of $\mathcal{O}\left(\ln(t)/\sqrt{t}\right)$ assuming a diminishing learning rate and where $t$ denotes the number of iterations. Numerical results verify our theoretical findings and demonstrate that our method reduces communication costs by approximately $75\%$ when compared to the state-of-the-art method.

## 1 Introduction

With the growing prevalence of computationally capable edge devices, there is a necessity for efficient learning algorithms that preserve data locality and privacy. One popular approach is decentralized ML. Under this regime, nodes within the network learn a global model by iterative model communication, whilst preserving data locality (Lalitha et al., 2018). It has been shown that decentralized ML algorithms can achieve similar accuracy and convergence rates to their centralized counterparts under certain network connectivity conditions (Nedic & Ozdaglar, 2009; Scaman et al., 2017; Koloskova et al., 2019; Lian et al., 2017). While decentralized ML eliminates the need for data uploading, (Bonawitz et al., 2019; Van Berkel, 2009; Li et al., 2020) observed that iterative model communication over rate-constrained channels creates a bottleneck. Particularly, the performance of decentralized training of large-scale deep neural network (DNN) models is substantially limited by excessively high communication costs (Chilimbi et al., 2014; Seide et al., 2014; Ström, 2015). This emphasizes the importance of designing efficient protocols for the communication and aggregation of high-dimensional local models.

Recent research has shown that efficiency of model communication can be enhanced by model sparsification and gradient/model compression. Model sparsification involves reducing the dimensionality of model parameters by exploiting their sparsity (Tan et al., 2011). On the other hand, model and gradient compression for decentralized ML has been studied in Alistarh et al. (2017); Koloskova et al. (2021; 2019); Han et al. (2015); Wen et al. (2017); Lin et al. (2018); Seide et al. (2014). It typically leverages lossy compression methods, such as quantization and source coding, to compress local model updates prior to communication. While this approach reduces communication bandwidth, it inevitably introduces imprecision during model aggregation.

**Related Work**. Kempe et al. (2003); Xiao & Boyd (2004); Boyd et al. (2006); Dimakis et al. (2010) proposed gossiping algorithms for consensus aggregation of local optimization results by leveraging peer-to-peer or neighborhood communication. Recently, Koloskova et al. (2019); Zhang et al. (2017); Scaman et al. (2017) adopted the gossiping algorithm for decentralized ML with convex and smooth loss functions, where convergence is guaranteed with constant and diminishing step sizes.

Continuing this work, Koloskova et al. (2021); Lian et al. (2017); Nadiradze et al. (2021); Tang et al. (2018) generalized the approach to smooth but non-convex loss functions and provided convergence guarantees. Furthermore, (Koloskova et al., 2019) and (Tang et al., 2018) utilized gradient compression by compressing the model updates before communication and aggregation. Notably, Alistarh et al. (2017) proposed a dithering-based quantization scheme as well as a symbol-wise source coding method for ML model compression. Meanwhile, Nadiradze et al. (2021) analyzed the convergence of asynchronous decentralized optimization with model compression. While communication compression is advantageous in of itself, Alistarh et al. (2018); Stich et al. (2018) analytically and numerically demonstrated the bandwidth gains from gradient sparsfication.

Fukushima & Mine (1981) proposed proximal gradient descent (PGD) for the optimization of non-smooth functions with separable convex and smooth components. Subsequently, Beck & Teboulle (2009) studied the convergence of PGD when the proximal optimization step can be executed via a soft-thresholding operation. Continuing this work, Schmidt et al. (2011) generalized the method for a convex objective with inexact gradient descent steps, while Sra (2012) established the convergence rate of inexact PGD for non-convex objectives. Meanwhile, in the decentralized convex optimization setting, Chen (2012) proved the convergence of inexact PGD with sub-optimal proximal optimization. Furthermore, Zeng & Yin (2018b) introduced general decentralized PGD algorithms and established the convergence rates for both constant and decreasing step sizes. However, to the best of our knowledge, the perturbations caused by data sub-sampling alongside stochastic gradient descent (SGD) updates and the lossy gradient compression have been overlooked in the literature.

**Our Contributions**. In this work, we address the challenge of enhancing the communication efficiency in compressed decentralized ML with a non-convex loss. We introduce the **M**ulti-**A**gent **L**earning via **Com**pressed updates for **P**roximal **S**tochastic **G**radient **D**escent (MALCOM-PSGD) algorithm, which communicates compressed and coded model updates. Our algorithm leverages model sparsification to reduce the model dimension by incorporating $\ell_1$ regularization during the SGD update. However, this non-smooth regularization term complicates the direct use of existing decentralized SGD algorithms in local training. In response, we introduce a strategy that combines decentralized proximal SGD to solve the regularized decentralized optimization, alongside gradient compression to alleviate the communication cost. Our contributions are summarized as follows.

- We introduce MALCOM-PSGD which utilizes gradient compression, vector source encoding, strategic aggregation and model sparsification to reduce the communication costs for decentralized learning. Through analysis and numerical evaluation we show that the amalgamation of these techniques enhances communication efficiency beyond the sum of their individual contributions. Numerically, we demonstrate a 75% bit rate reduction when compare to the state of the art.
- We prove that MALCOM-PSGD converges in the objective value for non-convex and smooth loss functions with a given compression error. MALCOM-PSGD exhibits a convergence rate of $\mathcal{O}(\ln t/\sqrt{t})$ and a consensus rate of $\mathcal{O}(1/t)$ with diminishing learning rates and consensus stepsizes, where $t$ represents the number of training rounds. Our aggregation scheme improves the accuracy but complicates the analysis because it does not preserve the average of the iterates. Combined with the non-smooth objective, we cannot directly use the proofs in Koloskova et al. (2021) and Zeng & Yin (2018b) to ensure convergece of MALCOM-PSGD.
- We employ the vector source coding scheme from Woldemariam et al. (2023) to minimize communication requirements, by encoding the support vector of entries that match a certain quantization level. Our use of compressed model updates and differential encoding allows us to reasonably assume we are creating a structure within our updates that this encoding scheme is most advantageous under. This allows us to benefit from the compression under this scheme.

## 2 DECENTRALIZED LEARNING

We consider a decentralized learning system composed of $n$ nodes, whose goal is to collaboratively minimize an empirical loss function for a neural network model, as

$$\min_{\boldsymbol{x}} \frac{1}{n} \sum_{i=1}^{n} F_i(\boldsymbol{x}; \mathcal{D}_i), \quad F_i(\boldsymbol{x}) := \frac{1}{|D_i|} \sum_{j=1}^{|D_i|} f_i(\boldsymbol{x}; \xi_{i,j}), \tag{1}$$

where $\boldsymbol{x} \in \mathbb{R}^d$ represents the model parameters, $F_i(\cdot; \mathcal{D}_i)$ is the local loss on node $i$ with respect to (w.r.t.) the local dataset $\mathcal{D}_i$, and $f_i(\cdot; \xi_{i,j})$ represents the loss w.r.t. the data sample $\xi_{i,j}$. We assume

that $F_i$ is finite-sum, smooth, and non-convex, which represents a broad range of machine learning applications, such as logistic regression, support vector machine, DNNs, etc.

**Communication**. At each iteration of the algorithm, nodes individually update local models using the proximal gradient of the cost obtained with a mini-batch of their own data sets. Then, they communicate the update with their neighbors according to a network topology specified by an undirected graph $\mathcal{G}(\{1, \cdots, n\}, E)$, where $E$ denotes the edge set representing existing communication links among the nodes. Let $\mathcal{N}_i := \{j \in V : (i, j) \in E\}$ denote the neighborhood of $i$. We define a mixing matrix $\boldsymbol{W} \in \mathbb{R}^{n \times n}$, with the $(i, j)-$th entry $w_{ij}$ denoting the weight of the edge between $i$ and $j$. We make the following standard assumptions about $\boldsymbol{W}$:

**Assumption 1** (The mixing matrix). *The mixing matrix $\boldsymbol{W}$ satisfies the following conditions:*

  i. *$w_{ij} > 0$ if and only if there exists an edge between nodes $i$ and $j$.*
 ii. *The underlying graph is undirected, implying that $\boldsymbol{W}$ is symmetric.*
iii. *The rows and columns of $\boldsymbol{W}$ have a sum equal to one, i.e., it is doubly-stochastic.*
 iv. *$\mathcal{G}(\{1, \cdots, n\}, E)$ is strongly connected, implying that $\boldsymbol{W}$ is irreducible.*

We align the eigenvalues of $\boldsymbol{W}$ in descending order of magnitude as $|\lambda_1| = 1 > |\lambda_2| \geq \cdots \geq |\lambda_n|$. In practice, the weights in $\boldsymbol{W}$ can be chosen by multiple metrics, such as node degrees, link distances, and communication channel conditions (Dimakis et al., 2010). This is the standard assumption for the mixing/gossiping matrix and is consistent with Koloskova et al. (2021).

**Model Sparsification**. As shown in (Chilimbi et al., 2014; Seide et al., 2014; Ström, 2015), the frequent model communication is regarded as the bottleneck in the decentralized optimization of (1), especially when optimizing large-scale ML models, such as DNNs. To facilitate the compression of the models updates, we promote model sparsity by adding $\ell_1$ regularization to the objective. Specifically, we replace the original objective in (1) with the following problem:

$$\min_{\boldsymbol{x} \in \mathbb{R}^d} \left\{ \mathcal{F}(\boldsymbol{x}) := \frac{1}{n} \sum_{i=1}^{n} F_i(\boldsymbol{x}; \mathcal{D}_i) + \mu \|\boldsymbol{x}\|_1 \right\}, \tag{2}$$

where $\mu > 0$ is a predefined penalty parameter. The $\ell_1$ regularization is an effective convex surrogate for the $\ell_0$-norm sparsity function, and thus is established as a computationally efficient approach for promoting sparsity in the fields of compression and sparse coding (Donoho, 2006). However, we note that adding $\ell_1$ regularization makes the objective in (2) non-smooth.[1]

**Preliminaries on Decentralized Optimization**. Decentralized optimization algorithms solve (2) by an alternation of local optimization and consensus aggregation. Specifically, defining $\boldsymbol{x}_i$ as the local model parameters in node $i$ for $\forall 1 \leq i \leq n$, (2) can be recast as

$$\min_{\boldsymbol{x}_i \in \mathbb{R}^d, \forall i} \quad \frac{1}{n} \sum_{i=1}^{n} (F_i(\boldsymbol{x}_i; \mathcal{D}_i) + \mu \|\boldsymbol{x}_i\|_1) \quad \text{subject to} \quad \boldsymbol{x}_i = \boldsymbol{x}_j, \forall (i, j) \in E. \tag{3}$$

The nodes iteratively update the local solutions $\{\boldsymbol{x}_i\}$ throughout $T$ iterations. Denote the $i$-th local model in iteration $t$ as $\boldsymbol{x}_i^{(t)}$. In iteration $t + 1$, each node $i$ first updates a local solution, denoted by $\boldsymbol{x}_i^{(t+\frac{1}{2})}$, by minimizing $\mathcal{F}_i(\boldsymbol{x}_i)$ using the local dataset and the preceding solution $\boldsymbol{x}_i^{(t)}$. Once every node completes its local update, they communicate with their neighbors and aggregate the local models by the following scheme.

$$\boldsymbol{x}_i^{(t+1)} = \boldsymbol{x}_i^{(t+1/2)} + \gamma_t \sum_{j=1}^{n} w_{i,j} \left( \boldsymbol{y}_j^{(t)} - \boldsymbol{x}_i^{(t+1/2)} \right) = (1 - \gamma_t) \boldsymbol{x}_i^{(t+1/2)} + \gamma_t \sum_{j \in \mathcal{N}_i}^{n} w_{ij} \boldsymbol{y}_j^{(t)}, \quad (4)$$

Here, $\gamma_t$ is the consensus step size (i.e., consensus "learning rate") and $\boldsymbol{y}_j^{(t)}$ is the inexact reconstruction of $\boldsymbol{x}_j$ using the compressed information node $j$ shared with node $i$. This aggregation protocol is non-standard and is thus one of our contributions. For comparison, using our notation, Koloskova et al. (2021)'s aggregation scheme is $\boldsymbol{x}_i^{(t+1)} = \boldsymbol{x}_i^{(t+1/2)} + \gamma_t \sum_{j=1}^{n} w_{i,j} \left( \boldsymbol{y}_j^{(t)} - \boldsymbol{y}_i^{(t)} \right)$

---

[1] Since our optimization design is specifically aimed at minimizing the regularized loss, $\mathcal{F}$, our analysis will predominantly concentrate on the convergence with respect to $\mathcal{F}$, rather than the unregularized loss $F$.

---

**Algorithm 1** MALCOM-PSGD

---

**Initialize:** $\boldsymbol{x}_i^{(0)}, \boldsymbol{y}_i^{(-1)} = \mathbf{0} \in \mathbb{R}^d, \forall 1 \leq i \leq n$.
1: **for** $t \in [0, \ldots, T-1]$ **do**                                    ▷ All nodes $i$ do in parallel
2:     $\boldsymbol{x}_i^{(t+1/2)} = \boldsymbol{x}_i^{(t)} - \eta_t \nabla F_i(\boldsymbol{x}_i^{(t)}, \xi_i^{(t)})$
3:     $\boldsymbol{q}_i^{(t)} = Q(\boldsymbol{x}_i^{(t+1/2)} - \boldsymbol{y}_i^{(t-1)})$                 ▷ Residual quantization; see (6).
4:     **for** $j \in \mathcal{N}_i$ **do**
5:         **Encode and send** $\boldsymbol{q}_i^{(t)}$     ▷ Residual communication with source encoding from Alg. 3.
6:         **Receive and decode** $\boldsymbol{q}_j^{(t)}$                              ▷ Receiver decoding.
7:         $\boldsymbol{y}_j^{(t)} = \boldsymbol{q}_j^{(t)} + \boldsymbol{y}_j^{(t-1)}$
8:     $\boldsymbol{z}_i^{(t+1)} = \boldsymbol{x}_i^{(t+1/2)} + \gamma_t \sum_{j \neq i} w_{i,j} \left( \boldsymbol{y}_j^{(t)} - \boldsymbol{x}_i^{(t+1/2)} \right)$     ▷ Consensus aggregation; cf. (4).
9:     $\boldsymbol{x}_i^{(t+1)} = \mathcal{S}_{\eta_t \mu} \left( \boldsymbol{z}_i^{(t+1)} \right)$          ▷ Proximal optimization by soft-thresholding; see (7).

---

**Objective of This Work.** First, we seek an algorithm for optimizing (2) that also incorporates quantized model updates. Second, we design a communication protocol that effectively compresses model updates to minimize the number of bits needed for the gossiping aggregation in (4).

## 3 PROPOSED ALGORITHM

MALCOM-PSGD solves (2) using decentralized proximal SGD with five major steps: *SGD*, *residual compression*, *communication and encoding*, *consensus aggregation*, and *proximal optimization*. We refer to Algorithm 1 for the illustration of the steps. Unless it is stated otherwise, we consider a synchronous decentralized learning framework over a static network for the sake of the analysis. However, MALCOM-PSGD can be readily extended to asynchronous and time-varying networks while maintaining comparable performance, as demonstrated numerically in Section B.4. The steps of MALCOM-PSGD are detailed below.

**SGD.** At iteration $t$, each node $i$ performs the mini-batch SGD update w.r.t. the gradient of the local empirical loss $F_i(\boldsymbol{x}; \mathcal{D}_i)$ with input $\boldsymbol{x}_i^{(t)}$. The updated local model is denoted by $\boldsymbol{x}_i^{(t+1/2)}$ in Step 2 of Alg. 1, where $\xi_i^{(t)}$ is the mini-batch data sampled from $\mathcal{D}_i$.

**Residual Quantization.** We directly utilize Assumption 1 from Koloskova et al. (2019) to define our compression operator $Q(\boldsymbol{x})$.

**Assumption 2.** *For any input $\boldsymbol{x}$, $Q(\boldsymbol{x}) : \mathbb{R}^d \rightarrow \mathbb{R}^d$ satisfies that*

$$\mathbb{E}[\|Q(\mathbf{x}) - \mathbf{x}\|^2] \leq (1 - \frac{1}{\tau})\|\mathbf{x}\|^2, \qquad Q(\mathbf{0}) = \mathbf{0}, \tag{5}$$

*where $\tau \in (0, 1)$ is a constant representing the expected compression error.*

Assumption 2 models a broad range of popular gradient compression schemes including QSGD (Alistarh et al., 2017), top$_k$ and rand$_k$ (Stich et al., 2018), and rescaled unbiased estimators. While any of the above compression schemes yield the same analytical results for consensus and convergence (Theorem 1 and Theorem 2), we introduce the following compression scheme for bit rate analysis and implementation. We integrate the uniform quantization scheme QSGD with uniform dithering and adaptive normalization as follows.

$$Q(x_i) = \mathbf{1}_{x_i \neq 0} \frac{1}{\tau} \left( \zeta \left( \frac{x_i - \min(\boldsymbol{x})}{\max(\boldsymbol{x}) - \min(\boldsymbol{x})}, \Gamma \right) \right), \; \zeta(x, \Gamma) = \frac{1}{\Gamma} \lfloor x\Gamma + u \rfloor, \tag{6}$$

where $\tau = 1 + d/\Gamma^2$, $x_i$ denotes the $i$-th entry of the input vector $\boldsymbol{x}$, $\mathbf{1}_A$ is the indicator function, $\zeta(x, \Gamma)$ is the uniform scalar quantizer with $\Gamma$ quantization levels, and $u$ is drawn from the uniform distribution over $[0, 1]$. We show in Appendix A.1 that the quantizer in (6), combined with the corresponding de-normalization process at the receiver, satisfies Assumption 2. We apply $Q(\cdot)$ element-wise to quantize the model differential $\boldsymbol{x}_i^{(t+\frac{1}{2})} - \boldsymbol{y}_i^{(t-1)}$ for each $i$ in the network. Here, the input values are adaptively normalized into the range $[0, 1]$ by the extreme values of the input vector.

We emphasize the importance of this adaptive normalization process in limiting the quantization error during the training, particularly when dealing with small input values.

**Source Coding**. The proximal operation promotes sparsity in the local models, encouraging shrinkage in the quantized model residuals, and thus motivating us to use fewer bits for fixed precision. This is accomplished by encoding the frequencies and positions of non-zero values over the support of each $\mathbf{q}_i^{(t)}$ by using a vector source encoder. We employ the source coding scheme proposed in Woldemariam et al. (2023), where the details are discussed in Algorithm 3 of Appendix A.3. The scheme is inspired by the notion of encoding the support of a sparse input. If the quantized coefficients that are non-zero tend to be concentrated around a limited number of modes, we can encode the support of those coefficients for each level efficiently as we expect most quantization levels to appear at a lower frequency. For notational simplicity, we omit the indices $i$ and $t$ from the quantized encoding input vector $\mathbf{q}_i^{(t)}$, representing it as $\mathbf{q}$ in the sequel. The type vector $\boldsymbol{t}(\mathbf{q})$ storing the frequency information of each level, denoted by $\chi_\ell, 0 \leq \ell \leq \Gamma - 1$, is first created, where the $\ell$-th type is $t_\ell(\mathbf{q}) = \sum_{j=1}^d \delta(q_j - \chi_\ell)$. Each entry of the type vector has an associated support vector $s_\ell$ denoting the positions in $\mathbf{q}$ that are equal to $s_\ell$, i.e., $s_\ell[j] = 1$ if $q_j = \chi_\ell$ and $s_\ell[j] = 0$ otherwise. Because $\sum_{\ell=0}^{\Gamma-1} s_\ell[j] = 1, \forall 1 \leq j \leq d$, for a support vector $s_\ell$ with $s_\ell[j] = 1$, we know that all subsequent support vectors for levels $\ell' > \ell$ have $s_{\ell'}[j] = 0$. Thus, these subsequent support vectors do not need to encode positional information for the values $\mathcal{I}_\ell$, the set of indices where $s_\ell[i] = 1$. Let $s_\ell[\mathcal{I}_\ell]$ denote the indices within $s_\ell$ that are communicated. It follows from Woldemariam et al. (2023) that $\mathcal{I}_\ell = \mathcal{I}_0 \setminus \bigcup_{\ell' < \ell} \mathcal{I}_{\ell'}$ where $\mathcal{I}_0 = \{1, \ldots, d\}$. The run-lengths within the support vectors $s_\ell[\mathcal{I}_\ell]$ are then encoded with Golomb encoding. For further details see Appendix A.3.

**Consensus Aggregation.** Each node $i$ aggregates the local models from its neighbors with its updated local model $\boldsymbol{x}_i^{(t+1/2)}$ (Step 9 of Alg. 1), leading to the aggregated model denoted by $\mathbf{z}_i^{(t+1)}$. The proposed consensus process differs from Koloskova et al. (2019), since we employ the true local model $\boldsymbol{x}_i^{(t+1/2)}$, rather than the reconstructed one $\boldsymbol{y}_i^{(t)}$, for aggregation. Even though it does not preserve the average of the iterates and complicates the analysis, this aggregation scheme was chosen because it reduces the error accumulation caused by $Q(\boldsymbol{x})$. Furthermore, this approach allows us to balance consensus and local training by adjusting the stepsize $\gamma_t$. In addition, it effectively reduces the consensus error, as shown in Section 4.

**Proximal Optimization.** To tackle the non-smoothness of the objective function in (2), we adopt the proximal SGD method, which decomposes (2) into a smooth component $F_i(\boldsymbol{x})$ and a convex but non-smooth component $\mu\|\boldsymbol{x}\|_1$. Then, $\boldsymbol{x}$ is updated by the SGD method as previously described and is subsequently combined with neighboring estimates during the consensus step. Finally, the model $\boldsymbol{x}^{(t+1)}$ is computed by applying a proximal operation w.r.t. $\mu\|\boldsymbol{x}\|_1$. This operation is characterized by a closed-form update expression derived from the soft-thresholding function. Specifically, let $\mathcal{S}_{\eta_t\mu}(x) = \max\{(|x| - \mu\eta_t), 0\}\operatorname{sign}(x)$ denote the soft-thresholding function. The proximal update step is given by

$$\boldsymbol{x}_i^{(t+1)} = \operatorname{prox}_{\eta_t, \mu\|\cdot\|}\left(\boldsymbol{z}_i^{(t+1)}\right) = \mathcal{S}_{\eta_t\mu}\left(\boldsymbol{z}_i^{(t+1)}\right), \tag{7}$$

where $\operatorname{prox}_{\eta_t, \mu\|\cdot\|}(\boldsymbol{z}) = \arg\min_{\boldsymbol{u}}\{\mu\eta_t\|\boldsymbol{u}\|_1 + \|\boldsymbol{u} - \boldsymbol{z}\|^2/2\}$ is the proximal operator. The soft-thresholding operation promotes model sparsity by truncating values with a magnitude less than $\mu\eta_t$. This step is important in accelerating convergence and conserving communication bandwidth.

**Remark.** MALCOM-PSGD aggregates the local models $\boldsymbol{z}_i^{(t+1)}$, which are updated in the SGD steps, *before* the proximal operation. This design allows us to model the compression error and the SGD update variance as perturbations to the proximal step. We analyze the impact of these perturbations on the convergence of MALCOM-PSGD in Section 4.

## 4 CONVERGENCE ANALYSIS

In this section, we analyze the convergence conditions of MALCOM-PSGD in terms of 1) the convergence of the consensus in model aggregation, and 2) the convergence of the optimization solution to (2). We denote the average of local models in round $t$ by $\bar{\boldsymbol{x}}^{(t)} = \frac{1}{n}\sum_{i=1}^n \boldsymbol{x}_i^{(t)}$. For ease of notation, we denote the model parameters in the matrix form by stacking the local models by column

as $\boldsymbol{X}^{(t)} := [\boldsymbol{x}_1^{(t)}, \cdots, \boldsymbol{x}_n^{(t)}]$ and $\overline{\boldsymbol{X}}^{(t)} := [\overline{\boldsymbol{x}}^{(t)}, \cdots, \overline{\boldsymbol{x}}^{(t)}]$. We impose the following assumptions on the training loss function, which is standard in the stochastic optimization literature.

**Assumption 3.** *Each local empirical loss function, i.e., $F_i$ in (1), satisfies the following conditions.*

i. *Each $\boldsymbol{x}_i \mapsto F_i$ is Lipschitz smooth with constant $L_i$. As a result, the sum $\sum_i F_i$ is Lipschitz smooth with constant $L = \max_i L_i$.*

ii. *Each $F_i(\boldsymbol{x}_i) + \mu \|\boldsymbol{x}_i\|_1$ is proper, lower semi-continuous, bounded below, and coercive[2].*

iii. *All the full batch gradient vectors are bounded above by $G < \infty$, i.e., $\|\nabla F_i(\boldsymbol{x}_i^{(t)})\|^2 \leq G^2, \forall i, t$, Moreover, the mini-batch stochastic gradient vectors are unbiased with bounded variance, i.e.,*

$$\mathbb{E}[\nabla F_i(\boldsymbol{x}_i^{(t)}; \xi_i^{(t)})] = \nabla F_i(\boldsymbol{x}_i^{(t)}), \quad \mathbb{E}[\|\nabla F_i(\boldsymbol{x}, \xi_i) - \nabla F_i(\boldsymbol{x})\|^2] \leq \sigma_i^2, \forall i, t. \quad (8)$$

By the AM-GM inequality, (8) implies

$$\sum_{i=1}^n \mathbb{E}[\|\nabla F_i(\boldsymbol{x}, \xi_i)\|^2] \leq 2 \sum_{i=1}^n \left( \mathbb{E}[\|\nabla F_i(\boldsymbol{x}, \xi_i) - \nabla F_i(\boldsymbol{x})\|^2] + \|\nabla F_i(\boldsymbol{x})\|^2 \right) \leq 2n(G^2 + \sigma^2),$$
$$(9)$$

where $\sigma^2 \triangleq \frac{1}{n} \sum_{i=1}^n \sigma_i^2$ is the average gradient variance. In (9), the term $\sigma^2$ measures the inexactness introduced to the mini-batch SGD step.

In this section, we consider a diminishing sublinear learning rate as seen in (Zeng & Yin, 2018b):

$$\eta_t = \frac{1}{L(t+a)^\epsilon}, \quad (10)$$

where $a \geq 1$ and $\epsilon \in (0, 1]$ are predefined hyperparameters controlling the decaying rate. Furthermore, we also choose a decreasing consensus stepsize surrogated by $\eta_t$, as

$$\gamma_0 \leq \frac{1 - a^{-\epsilon}}{1 - \lambda_n}, \qquad \gamma_t \leq \eta_t, \ \gamma_{t+1} \leq \gamma_t, \forall t. \quad (11)$$

We first show in the following that the consensus error converges to zero in MALCOM-PSGD.

**Theorem 1.** *Suppose Assumptions 1-3 hold. Let $\eta_t$ and $\gamma_t$ be defined from 10 and 11 and define $\omega = \frac{(1-|\lambda_2|)^2}{8\tau}$, where $\tau$ is from Assumption 2. Suppose the following conditions hold.*

$$\gamma_t \leq \frac{1 - |\lambda_2|}{4\tau}; \qquad a \geq \frac{8\epsilon}{\omega}; \qquad \boldsymbol{x}_i^{(0)} = \boldsymbol{0}, \forall i. \quad (12)$$

*Then, for $\forall t > 0$, $\forall \epsilon \in (0, 1]$, and $\forall Q(\bullet)$ that satisfies Assumption 2 we have*

$$\sum_{i=1}^n \mathbb{E}\|\boldsymbol{x}_i^{(t)} - \overline{\boldsymbol{x}}^{(t)}\|^2 \leq \frac{C}{\omega^3}\left(2G^2 + 2\sigma^2 + \frac{2}{3}\mu^2 d\right)n\eta_t^2, \quad (13)$$

*where $C < 116$ is an independent constant. Furthermore, applying the value of $\eta_t$ in (10), the consensus error $\sum_{i=1}^n \mathbb{E}\|\boldsymbol{x}_i^{(t)} - \overline{\boldsymbol{x}}^{(t)}\|^2$ converges to zero on the rate of $\mathcal{O}(1/t)$.*

For the proof of Theorem 1, we refer to Appendix A.4. Theorem 1 is applicable to a broad range of compression and coding schemes that satisfy Assumption 2. While the mini-batch sampling variance in the SGD step is characterized by $\sigma$, a more significant compression error leads to a larger $\tau$, which in turn is manifested by a decreased $\omega$ in (13). Theorem 1 indicates a consensus rate of $\mathcal{O}(1/t)$ which asymptotically matches (Koloskova et al., 2021, Lemma A.2). However, we require a diminishing learning rate which is a stricter condition and is required because of the non-smoothness induced by the regularization term. Additionally, because our aggregation scheme does not preserve the average, the proof in Koloskova et al. (2021) does not hold for MALCOM-PSGD.

Define $\mathcal{F}(\boldsymbol{X}) = 1/n \sum_{i=1}^n (F_i(\boldsymbol{x}_i) + \mu \|\boldsymbol{x}_i\|_1)$. Now we detail the final assumptions required for proving the convergence of MALCOM-PSGD.

**Assumption 4.** *Let $\{\|\boldsymbol{X}^{(t)}\|_F\}_{t=0}^\infty$ be the local models' sequence obtained by Algorithm 1 within the training iterations. The corresponding objective values $\{\mathcal{F}(\boldsymbol{X}^{(t)})\}_{t=0}^\infty$ are always finite. Together with the coercivity of $\mathcal{F}$ in Assumption 3(ii), it follows that the norms of $\{\boldsymbol{X}^{(t)}\}_{t=0}^\infty$ are bounded as $\|\boldsymbol{X}^{(t)}\|_F \leq B, \forall t$, for some constant $B < \infty$.*

---

[2]A function $h(\boldsymbol{x})$ is coercive if $\|\boldsymbol{x}\| \to \infty$ implies $h(\boldsymbol{x}) \to \infty$.

Assumption 3(ii) is consistent with Zeng & Yin (2018b) and under this condition, Assumption 4 requires that the loss values produced in training iterations of our algorithm are finite. This assumption can be reliably met with practical machine learning solvers. We note that Assumption 4 is not required in Koloskova et al. (2021) and Zeng & Yin (2018b), but is critical for our analysis since we need it to bound the optimality gap. Koloskova et al. (2021) is able to avoid this assumption by assuming a smooth objective function while Zeng & Yin (2018b) avoids it by assuming loss-less communication.

**Theorem 2.** *Let the weighted average objective value be*

$$\overline{\mathcal{F}}_t = \frac{\sum_{k=0}^t \eta_k \mathcal{F}(\overline{\boldsymbol{X}}^{(k+1)})}{\sum_k \eta_k}. \tag{14}$$

*Let $\mathcal{F}^*$ be the optimal objective value to $\mathcal{F}(\boldsymbol{X})$. With Assumptions 1-4 and (12), $\overline{\mathcal{F}}_t$ satisfies*

$$\mathbb{E}\left[\overline{\mathcal{F}}_t - \mathcal{F}^\star\right] \leq \left(C_1 + \frac{C_2}{\sqrt{n}}\right) \frac{\sum_{k=0}^t \eta_k^2}{\sum_{k=0}^t \eta_k} + \frac{C_3}{\sum_{k=0}^t \eta_k}. \tag{15}$$

*Here, $C_1$, $C_2$, and $C_3 \sim \mathcal{O}(1)$ are some constants independent to $t$ and $n$ on the order of such that*

$$C_1 \leq \frac{116}{2\omega^3}\left(2G^2 + 2\sigma^2 + \mu^2 d\right) + \|\boldsymbol{x}^\star\|_2 \sqrt{\frac{116}{2\omega^3}\left(2G^2 + 2\sigma^2 + \mu^2 d\right)},$$

$$C_2 \leq B\sqrt{\frac{116}{2\omega^3}\left(2G^2 + 2\sigma^2 + \mu^2 d\right)},$$

$$C_3 = \|\boldsymbol{x}^*\|_2^2/2,$$

*where $\omega$ is defined in Theorem 1, and $B$ is the constant defined in Assumption 4. Furthermore, the convergence rate in (15) can be determined by choice of $\epsilon$ in (10). In particular, when $\epsilon = \frac{1}{2}$, $\mathbb{E}\left[\overline{\mathcal{F}}_t\right]$ converges to $\mathcal{F}^\star$ on the highest rate of*

$$\mathbb{E}\left[\overline{\mathcal{F}}_t - \mathcal{F}^\star\right] = \mathcal{O}\left(\frac{\ln t}{\sqrt{t}} + \frac{\ln t}{\sqrt{nt}} + \frac{1}{\sqrt{t}}\right) = \mathcal{O}\left(\frac{\ln t}{\sqrt{t}}\right). \tag{16}$$

The proof of Theorem 2 can be found in Appendix A.5. Theorems 1 and 2 show a trade-off between consensus and convergence rates, depending on the value of $\epsilon$, as both the consensus step size $\gamma_t$ and the learning rate $\eta_t$ are influenced by $\epsilon$. In particular, Theorem 2 indicates that the optimal value for $\epsilon$ is $\epsilon = \frac{1}{2}$, leading to the convergence rate of $\mathcal{O}(\ln(t)/\sqrt{t})$. We note that MALCOM-PSGD exhibits the same convergence rate w.r.t. $t$ as the error-free decentralized PGD approach in Zeng & Yin (2018b). However, the quantization error in $\omega$ and the SGD variance $\sigma^2$ amplify the values of the multiplicative terms in $C_1$ and $C_2$, slowing down the convergence speed. For a decentralized learning network exhibiting higher connectivity, the eigenvalue $|\lambda_2|$ of the mixing matrix $\mathbf{W}$ is generally smaller. This results in a larger $\omega$ as per Theorem 1, consequently leading to improved consensus and convergence rates according to Theorems 1 and 2.

The convergence order $\mathcal{O}\left((1 + 1/\sqrt{n})\ln t/\sqrt{t}\right)$ stated in Theorem 2 indicates a diminishing speedup with an increase in the number of devices $n$. This rate is notably slower than the rate of $\mathcal{O}(\frac{1}{n\sqrt{t}})$ achieved by decentralized PSGD with a convex loss and error-free communication, as demonstrated in (Zeng & Yin, 2018a, Theorem 4(e)). The slower rate is due to the non-convex nature of the loss function in (1). Also, our analysis, focusing on the convergence of the non-smooth objective value $\mathcal{F}$ to its minimum, does not exhibit a linear speedup in terms of $n$ in Koloskova et al. (2021), which examines the convergence of the gradient vector of a smooth loss.

## 5 COMMUNICATION BIT RATE ANALYSIS

As the training converges, we expect a diminishing residual to be quantized in Step 3 of Alg. 1, resulting in a sparse quantized vector. Let $f^t(\cdot)$ be the probability mass function (PMF) of the quantization output $\mathbf{q}_i^{(t)}$ at iteration $t$, and denote the corresponding (re-ordered) frequency of the $\ell$-th quantization level, $0 \leq \ell \leq \Gamma - 1$, where $\{f_\ell^t\}_{\forall \ell}$ satisfies $f_\ell^t \geq f_{\ell+1}^t, \forall \ell$. The quantization mapping described in Section 3 scales an input vector according to its range, essentially shrinking the support of $f^t(\cdot)$ as $t$ grows, given that the precision is fixed. In equation (11) of Woldemariam

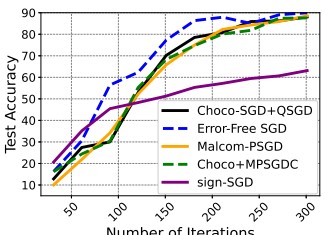 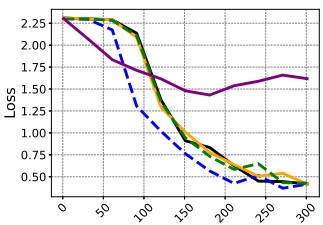

| Algorithms | Avg. (MB) | Std. Dev. |
|---|---|---|
| Maclcom | **12.74** | 0.090 |
| Error-Free | 241.1 | 0.000 |
| Choco+QSGD | 58.59 | 2.074 |
| Choco+OurComp | 14.85 | **0.066** |
| signSGD | 16.74 | 0.000 |

(a) Test accuracy per iterations.  (b) Training loss over iterations.  (c) Bits transmitted per iteration.

Figure 1: Synchronous FL with the MNIST dataset and a heterogeneous data distribution

et al. (2023) , it is shown that the bit length of encoding $\mathbf{q}_i^{(t)}$ is upper bounded by

$$d \left( H(f^t) + 2.914(1 - f_0^t) + f_0^t \log_2 f_0^t + \sum_{\ell=1}^{\Gamma-1} f_\ell^t \log_2(1 - \sum_{m=0}^{\ell-1} f_m^t) \right), \qquad (17)$$

where $f_0$ is the PMF associated with the most frequent quantization level, $H(f^t)$ is the entropy of $f^t$, and $d$ is the size of the model. By computing the empirical PMF $f^t$ in each iteration, (17) provides a formula for the required number of bits in each local model communication. As mentioned before, it is expected that the types will vary greatly as large values will rarely occur, which has been exploited to achieve compression gains in our compression scheme.

**Analysis Under Laplace Residual Model**. The computation of communication costs using (17) necessitates an understanding of the prior distribution of the quantization inputs, i.e.,, the local model residuals $\{\boldsymbol{x}_i^{(t+\frac{1}{2})} - \boldsymbol{y}_i^{(t-1)}\}$. The accurate joint distribution of these residuals is often unavailable prior to the training process. However, considering that the $\ell_1$-regularization in (2) promotes sparse local models, we can effectively approximate the residual distribution using a suitable sparse distribution model. For clarity in our explanation, we assume that at each iteration $t$, all the residuals follow an i.i.d. zero-mean Laplace distribution characterized by the time-varying Laplace diversity parameter $\rho_t$, where a sparser model residual leads to a smaller $\rho_t$. With a large $t$, we prove in Appendix A.3 that the communication bits for encoding each quantized model residual vector are bounded by:

$$(17) < d \left( \log_2(2e\rho_t) + 2.914 e^{-\frac{\ln(d/\epsilon)}{\Gamma}} \right) = \mathcal{O}\left( d \left( e^{-1/\Gamma} - \log t \right) \right), \qquad (18)$$

with probability at least $1 - \epsilon$ for small $\epsilon \in [0, 1]$.

(18) highlights that the communication efficiency of our compression scheme improves with an increased number of quantization levels $\Gamma$ on the order of $\mathcal{O}(de^{-1/\Gamma})$, which grows slower $\mathcal{O}(\log \Gamma)$. However, the advantage obtained from decreasing $\Gamma$ must be balanced against the degradation in convergence speed because a smaller $\Gamma$ also results in higher quantization error and thus slower convergence, as shown in (6) and Section 4. In summary, under the Laplace residual model, (18) shows that the required communication bits of our compression scheme converges to $\mathcal{O}(de^{-1/\Gamma})$ with a rate of $\mathcal{O}(-d \log t)$. We refer to Appendix A.3 for the analysis and comparison with other methods.

## 6 NUMERICAL RESULTS

In this section, we evaluate the performance of MALCOM-PSGD through simulations of decentralized learning tasks on image classification using the MNSIT (Deng, 2012) dataset distributed in a heterogeneous fashion consistent with Konečný et al. (2016). We compare MALCOM-PSGD with CHOCO-SGD (Koloskova et al., 2021) paired with the compression scheme of QSGD given in (Alistarh et al., 2017) and with CHOCO-SGD paired with the compression scheme utilized by MALCOM-PSGD. The details of the experimental set up can be found in Appendix B along with additional experiments. For the results presented here, models were trained for a fixed number of iterations and evaluated on the testing set while the number of bits were empirically computing using each Algorithm's respective encoding scheme. Monte Carlo simulations were performed for each experiment. Figure 1 showcases the testing accuracy, training loss, and number of bits per iteration over a fully connected network for MALCOM-PSGD, CHOCO-SGD+QSGD, CHOCO-SGD+Our

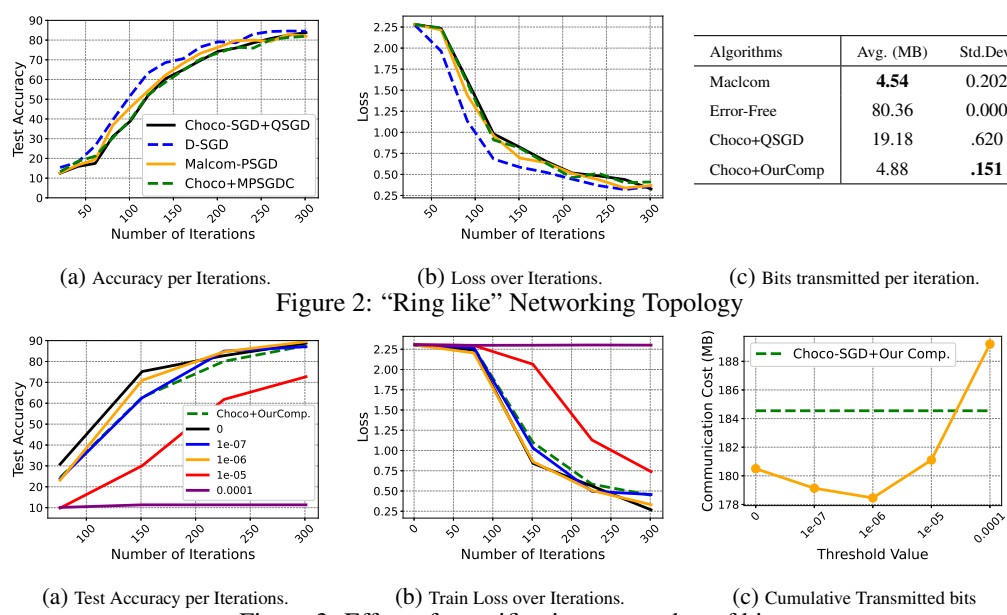

(a) Accuracy per Iterations.     (b) Loss over Iterations.     (c) Bits transmitted per iteration.

Figure 2: "Ring like" Networking Topology

(a) Test Accuracy per Iterations.     (b) Train Loss over Iterations.     (c) Cumulative Transmitted bits

Figure 3: Effect of sparsification on number of bits

Compression, signSGD (Bernstein et al., 2018), and the error free decentralized SGD. Since the baseline algorithms optimize the unregularized loss, we evaluate the performance of all the algorithms by *the unregularized loss F* for a uniform comparison. Similar to Figure 1, in Figure 2, we consider a "ring like"[3] network topology, and we compare the same algorithms, except for signSGD since it is only adapted to the Federated Learning case.

Figure 3 showcases the effects of different thresholding parameters ($\mu$). From Figure 1 and Figure 2, one can see that MALCOM-PSGD achieves equivalent test accuracy and convergence rate as CHOCO-SGD but we observe a 75% reduction in bits transmitted per iteration. When compared to signSGD in the Federated Learning scheme we achieve a 19% reduction. Lastly, Figure 3 demonstrates that the improvement in communication efficiency is the result of our aggregation technique, sparsification, and efficient encoding scheme. In Figure 3c, the x-axis is different thresholding parameters while the green line represents CHOCO-SGD with Our Compression scheme. Notice that when MALCOM-PSGD does not utilize sparsification it still uses fewer bits than CHOCO-SGD. This implies that our aggregation scheme, independent of sparsification, improves communication efficiency. Furthermore, Figure 3c indicates that model sparsification provides an additional improvement in communication efficiency, demonstrating that MALCOM-PSGD as a whole is more communication efficient than the sum of its parts.

## 7 CONCLUSION

We introduced the MALCOM-PSGD algorithm for decentralized learning with finite-sum, smooth, and non-convex loss functions. Our approach sparsifies local models by non-smooth $\ell_1$ regularization, rendering the implementation of the conventional SGD-based model updating challenging. To address this challenge, we adopted the decentralized proximal SGD method to minimize the regularized loss, where the residuals of local SGD updates are shared and aggregated prior to the local proximal optimization. Furthermore, we employed dithering-based gradient quantization and vector source coding schemes to compress model communication and leverage the low entropy of the updates to reduce the communication cost. By characterizing data sub-sampling and compression errors as perturbations in the proximal operation, we quantified the impact of gradient compression on training performance and established the convergence rate of MALCOM-PSGD with diminishing stepsizes. Moreover, we analyzed the communication cost in terms of the asymptotic code rate for the proposed algorithm. Numerical results validate the theoretical findings and demonstrate the improvement of our method in both learning performance and communication efficiency.

---

[3]We formally describe this network topology in Appendix B.1.1 including a diagram and mixing matrix.

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

# A APPENDIX

## A.1 THE QUANTIZATION SCHEME IN (6) SATISFIES ASSUMPTION 2

We denote the effective quantization scheme combining (6) and the de-normalization procedure at the receiver end by $\hat{Q}(\cdot)$. Specifically, for any input vector $\boldsymbol{x} \in \mathbb{R}^d$, the $i$-th entry of the output is given by $\hat{Q}(x_i) = (\max(\boldsymbol{x}) - \min(\boldsymbol{x}))Q(x_i) + \min(\boldsymbol{x})$. Let $h(x_i) = \dfrac{x_i - \min(\boldsymbol{x})}{\max(\boldsymbol{x}) - \min(\boldsymbol{x})}$, $r(\boldsymbol{x}) = \max(\boldsymbol{x}) - \min(\boldsymbol{x})$, and $m = \min(\boldsymbol{x})$. We have

$$\hat{Q}(x_i) = \frac{1}{\tau} \left( \zeta\left(h(x_i)\right) r(\boldsymbol{x}) + m \right).$$

From (Alistarh et al., 2017, Lemma 3.1), the operator $\zeta(\cdot)$ is unbiased, i.e., $\mathbb{E}[\zeta(h(\boldsymbol{x}))] = h(\boldsymbol{x})$, where the expectation is taken w.r.t. the uniform dithering process. Moreover, let $l$ be the quantization region of $x_i$ such that $h(x_i) \in [l/\Gamma, (l + 1/\Gamma)]$. Let $p_i$ denote the probability that $h(x_i)$ is mapped to $(l + 1)/\Gamma$, where we have $p_i = h(x_i)\Gamma - l$. In other words, we have $\zeta(h(x_i), \Gamma)$ equal to $(l + 1)/\Gamma$ with probability $p_i$ and equal to $(l + 1)/\Gamma$ with probability $1 - p_i$. Therefore, we have

$$\mathbb{E}[\zeta(h(x_i), \Gamma)^2] = \mathbb{E}[\zeta(h(x_i), \Gamma)]^2 + \mathrm{Var}(\zeta(h(x_i), \Gamma)) = h^2(x_i) + \frac{p(1-p)}{\Gamma^2} \leq h^2(x_i) + \frac{1}{4\Gamma^2},$$

where $\mathrm{Var}(\cdot)$ denotes the variance of the input random variable, and the last inequality follows from $p(1-p) \leq 1/4$.

Note that

$$\begin{aligned}
\mathbb{E}[\|\hat{Q}(\boldsymbol{x})\|^2] &= \frac{1}{\tau^2} \mathbb{E}\left[ \sum_{i=1}^{d} \left( r(\boldsymbol{x})\zeta(h(x_i), \Gamma) + m \right)^2 \right] \\
&= \frac{1}{\tau^2} \left( \sum_{i=1}^{d} \mathbb{E}\left[ r^2(\boldsymbol{x})\zeta(h(x_i), \Gamma)^2 \right] + \sum_{i=1}^{d} \mathbb{E}\left[ 2r(\boldsymbol{x})m\zeta(h(x_i), \Gamma) \right] + \sum_{i=1}^{d} m^2 \right) \\
&\leq \frac{1}{\tau^2} \left( r^2(\boldsymbol{x}) \sum_{i=1}^{d} \left( h^2(x_i) + \frac{1}{4\Gamma^2} \right) + 2r(\boldsymbol{x})m \sum_{i=1}^{d} h(x_i) + dm^2 \right).
\end{aligned} \tag{19}$$

For $h(\cdot)$, we have

$$\sum_{i=1}^{d} h(x_i) = \frac{1}{r(\boldsymbol{x})}\left(-dm + \sum_{i=1}^{d} x_i\right), \tag{20}$$

$$\sum_{i=1}^{d} h^2(x_i) = \frac{1}{r^2(\boldsymbol{x})} \sum_{i=1}^{d} x_i^2 - \frac{2m}{r^2(\boldsymbol{x})} \sum_{i=1}^{d} x_i + \frac{dm^2}{r^2(\boldsymbol{x})}. \tag{21}$$

---

**Algorithm 2** The matrix form of MALCOM-PSGD.

---

**Initialize:** $\boldsymbol{X}^{(0)}, \boldsymbol{Y}^{(-1)} = 0.$
1: **for** $t \in [0, \ldots, T-1]$ **do**           ▷ All nodes $i$ do in parallel
2:    $\boldsymbol{X}^{(t+1/2)} = \boldsymbol{X}^{(t)} - \eta_t \nabla \boldsymbol{F}\left(\boldsymbol{X}^{(t)}, \xi^{(t)}\right)$
3:    $\boldsymbol{Q}^{(t)} = Q(\boldsymbol{X}^{(t+1/2)} - \boldsymbol{Y}^{(t-1)})$       ▷ Residual quantization; see (6).
4:    **for** $j \in \mathcal{N}_i$ **do**
5:      **Encode and send** $\boldsymbol{Q}^{(t)}$   ▷ Residual communication with source encoding from Alg. 3.
6:      **Receive and decode** $\boldsymbol{Q}^{(t)}$         ▷ Receiver decoding.
7:      $\boldsymbol{Y}^{(t)} = \boldsymbol{Q}^{(t)} + \boldsymbol{Y}^{(t-1)}$
8:    $\boldsymbol{Z}^{(t+1)} = (1-\gamma_t)\boldsymbol{X}^{(t+1/2)} + \gamma_t \boldsymbol{Y}^{(t)}\boldsymbol{W}$    ▷ Consensus aggregation; cf. (4).
9:    $\boldsymbol{X}^{(t+1)} = \mathcal{S}_{\eta_t \mu}\left(\boldsymbol{Z}^{(t+1)}\right)$     ▷ Proximal optimization by soft-thresholding; see (7).

---

Substituting (20) and (21) into (19) we have:

$$\mathbb{E}[\|\hat{Q}(\boldsymbol{x})\|^2] = \frac{1}{\tau^2}\left(\|\boldsymbol{x}\|_2^2 + r^2(\boldsymbol{x})\frac{d}{4\Gamma^2}\right) \overset{(a)}{\leq} \frac{1}{\tau^2}\left(1 + \frac{d}{\Gamma^2}\right)\|\boldsymbol{x}\|_2^2 \overset{(a)}{=} \frac{\|\boldsymbol{x}\|_2^2}{\tau},$$

where $(a)$ follows from $r^2(\boldsymbol{x}) \leq 4\|\boldsymbol{x}\|_\infty^2 \leq 4\|\boldsymbol{x}\|_2^2$ and $(b)$ follows from the definition of $\tau$. Finally, we have

$$\mathbb{E}\left[\|\hat{Q}(\boldsymbol{x}) - \boldsymbol{x}\|^2\right] = \mathbb{E}\left[\sum_{i=1}^d \hat{Q}(x_i)^2\right] - 2\mathbb{E}\left[\sum_{i=1}^d \hat{Q}(x_i)x_i\right] + \mathbb{E}\left[\sum_{i=1}^d x_i^2\right]$$

$$\leq \frac{1}{\tau}\|\boldsymbol{x}\|^2 - \frac{2}{\tau}\|\boldsymbol{x}\|^2 + \|\boldsymbol{x}\|^2 = \left(1 - \frac{1}{\tau}\right)\|\boldsymbol{x}\|^2,$$

which completes the proof.

## A.2   MATRIX REPRESENTATION OF MALCOM-PSGD AND USEFUL LEMMAS

To facilitate our proofs, we recast MALCOM-PSGD into an equivalent matrix form. Specifically, we define $\boldsymbol{Y}^{(t)} = [\boldsymbol{y}_1^{(t)}, \cdots, \boldsymbol{y}_n^{(t)}]$, $\boldsymbol{Z}^{(t)} = [\boldsymbol{z}_1^{(t)}, \cdots, \boldsymbol{z}_n^{(t)}]$, $\boldsymbol{Q}^{(t)} = [\boldsymbol{q}_1^{(t)}, \cdots, \boldsymbol{q}_n^{(t)}]$, $\xi^{(t)} = [\xi_1^{(t)}, \cdots, \xi_n^{(t)}]^T$, and $\nabla F(\boldsymbol{X}^{(t)}, \xi^{(t)}) = [\nabla F_i(\boldsymbol{x}_1^{(t)}, \xi_1^{(t)}), \cdots, \nabla F_n(\boldsymbol{x}_n^{(t)}, \xi_n^{(t)})]$. Together with the matrices $\boldsymbol{X}^{(t)}$ and $\overline{\boldsymbol{X}}^{(t)}$ defined in Section 4, we have the matrix form of Alg. 1, as shown in Alg. 2.

We also provide two useful results for the proof.

**Lemma 1.** *For any* $\mathbf{A}, \mathbf{B}$ *with the same dimension, we have, for any* $\alpha > 0$*:*

$$\|\mathbf{A} + \mathbf{B}\|_F^2 \leq (1+\alpha)\|\mathbf{A}\|_F^2 + (1 + \alpha^{-1})\|\mathbf{B}\|_F^2, \tag{22}$$

$$\|\boldsymbol{AB}\|_F \leq \|\boldsymbol{A}\|_F \|\boldsymbol{B}\|_2. \tag{23}$$

**Lemma 2.** *For the mixing matrix* $\boldsymbol{W}$ *satisfying Assumption 1, we have, for any* $k > 0$,

$$\left\|\boldsymbol{W}^k - \frac{1}{n}\mathbf{1}\mathbf{1}^\top\right\|_2 \leq |\lambda_2|^k = (1-\delta)^k, \tag{24}$$

$$\|\mathbf{I} - \boldsymbol{W}\|_2^2 = \lambda_n^2, \tag{25}$$

*where we define* $\delta := 1 - |\lambda_2|$.

*Proof.* See (Koloskova et al., 2019, Lemma 16).              □

## A.3   FURTHER DISCUSSIONS ON THE ENCODING ALGORITHM

**Preliminaries on Source Encoding.** Elias coding Elias (1975) is a universal encoding scheme for unknown distributions that are assumed to generally have small integer values. Elias omega coding recursively encodes a value in binary: the string that encodes a value has the binary encoding of the value appended to itself, whose length becomes the subsequent value to be encoded in binary.

---

**Algorithm 3** The source encoding scheme.

---

1: **function** ENCODE($\boldsymbol{q}$, $\Gamma$)
2:     Compute $\boldsymbol{t}(\boldsymbol{q})$ and encode it with Elias omega encoding.
3:     Initialize $\mathcal{I} = \{1, \ldots, N\}$.
4:     **for** $\ell = 1, \ldots, \Gamma - 1$ **do**
5:         $\mathcal{I}' = \emptyset$, $\mathcal{R}_\ell = \emptyset$.
6:         Compute $M_\ell = [(\ln 2)(N - \sum_{m \leq \ell} t_m(\boldsymbol{q}))/t_\ell(\boldsymbol{q})]$.
7:         $r = 0$.
8:         **for** $j \in \mathcal{I}$ **do**
9:             **if** $\delta(q_j - \ell) = 0$ **then**
10:                $\mathcal{R}_\ell = \mathcal{R}_\ell \cup \{r\}$.
11:                **if** $|\mathcal{R}_\ell| < t_\ell(\boldsymbol{q})$ **then**
12:                     Encode $r$ by Golomb coding with parameter $M_\ell$.
13:                $r = 0$.
14:                $\mathcal{I}' = \mathcal{I}' \cup \{n\}$.
15:             **else**
16:                $r = r + 1$.
17:         $\mathcal{I} = \mathcal{I} \setminus \mathcal{I}'$.

---

Golomb coding Golomb (1966) an encoding scheme parameterized by an integer $M$ that divides the value to be encoded. The quotient and remainder are separately encoded in unary and binary, respectively.

**Algorithm Implementation**. The following algorithm describes the encoding scheme proposed in Woldemariam et al. (2023). In this scheme, the type vector is assumed to be in descending order, while implementation allows for an unordered type vector. In Algorithm 3, the notation $[x]$ denotes the nearest integer of $x$. To decode, the first $\Gamma$ strings encoded with Elias omega coding are decoded with Elias decoding to retrieve the type vector. The run-lengths are then decoded with Golomb decoding and used to iteratively reconstruct the support vectors. With the positional information encoded through the support vectors, the decoder can fill in values of $\chi_\ell$ for all $\ell$.

**Bit Analysis Under Laplace Inputs**. Calculating the number of communication bits using (**??**) requires the knowledge of the distribution of the local model residuals. Tracking the exact joint distribution of the local residuals $\boldsymbol{x}_i^{(t+\frac{1}{2})} - \boldsymbol{y}_i^{(t-1)}$ over iterations is generally challenging. As an alternative, we can fit the prior distribution of the residuals under a specific statistical model. Given that the $\ell_1$-minimization in (2) encourages sparsity in local model updates, a sparse prior distribution for the quantization input is a logical choice. For instance, we can assume the unquantized residuals $\{\boldsymbol{x}_i^{(t+\frac{1}{2})} - \boldsymbol{y}_i^{(t-1)}\}$ follow a time-varying i.i.d. Laplace distribution denoted as $p_t$. Specifically, each entry of the vector, denoted by $\tilde{q}_j^t, 1 \leq j \leq d$, follows the distribution of

$$p_t(\tilde{q}_j^t) = \text{Lap}(0, \rho_t) = \frac{1}{2\rho_t} e^{-|\tilde{q}_j^t|/\rho_t}, \tag{26}$$

where $\rho_t$ is the Laplace diversity parameter at iteration $t$. A smaller $\rho_t$ implies less variance in the value of $\tilde{q}_j^t$ and a sparser model residual for quantization.

Then, we bound the PMF $f_t$ of the quantized value $Q(\tilde{q}_j^t)$ by the entropy inequality $H(f^t) \leq h(p_t) = \log_2(2\rho_t e)$, where $h(p_t)$ represents the differential entropy of the distribution $p_t$. Furthermore, we have

$$f_0^t \overset{(a)}{\geq} \Pr(Q(\tilde{q}_j^t) = 0) = \Pr\left(-\frac{\Upsilon_t}{2\Gamma} \leq \tilde{q}_j^t \leq \frac{\Upsilon_t}{2\Gamma}\right)$$

$$\overset{(b)}{=} 2\Pr\left(0 \leq \tilde{q}_j^t \leq \frac{\Upsilon_t}{2\Gamma}\right) = 1 - e^{-\frac{\Upsilon_t}{2\Gamma\rho_t}}, \tag{27}$$

where $\Upsilon_t = \max(\boldsymbol{x}_i^{(t+\frac{1}{2})} - \boldsymbol{y}_i^{(t-1)}) - \min(\boldsymbol{x}_i^{(t+\frac{1}{2})} - \boldsymbol{y}_i^{(t-1)})$ is the time-varying range of the model residual vector used for adaptive normalization in (6), $(a)$ follows from that $f_0^t$ is the largest value in the PMF $f^t$, and $(b)$ follows from the symmetry of the Laplace distribution around zero. The next step is to bound the random variable $\Upsilon_t/\rho_t$. Using the property for the Laplace distribution

| Compression method | # of bits per local communication |
|---|---|
| Our scheme | $(17) < d(H(f^t) + \text{const}) \le d(\log \Gamma + \text{const})$ |
| QSGD | $d(\log \Gamma + \text{const})$ |
| signSGD | $d$ |

Table 1: Comparison of communication complexity with a large $t$ in the worst-case scenario.

$\Pr(\max_{1 \le j \le d} |q_j^t| > \vartheta \rho_t) \le de^{-\vartheta}, \forall \vartheta > 0$, and let $\epsilon = de^{-\vartheta}$, we have

$$\Pr\left(\max_{1 \le j \le d} |q_j^t| > (\ln d - \ln \epsilon)\rho_t\right) \le \epsilon. \tag{28}$$

Since $q_j^t$ is an i.i.d. continuous variable and the Laplce distribution is symmetric, we have $\Upsilon_t = 2\max_{1 \le j \le d} |q_j^t|$. Combining this result with (29), with probability at least $1 - \epsilon$ for a small $\epsilon < 1$, we have

$$f_0^t \ge 1 - e^{-\frac{\ln d - \ln \epsilon}{\Gamma}}. \tag{29}$$

Substituting this into the bit bound in (17), we have

# of bits per communication $\le d\big(H(f^t) + 2.914(1 - f_0^t) + f_0^t \log_2 f_0^t + \underbrace{\sum_{\ell=1}^{\Gamma-1} f_\ell^t \log_2(1 - \sum_{m=0}^{\ell-1} f_m^t)}_{<0}\big)$

$$< d(\log_2(2e\rho_t) + 2.914e^{-\frac{\ln d - \ln \epsilon}{\Gamma}}). \tag{30}$$

Notably, our compression scheme achieves smaller communication cost with sparser inputs (i.e., a smaller $\rho_t$). By using the convergence result in Theorem 1, we can bound the value of $\rho_t$ as follows.

**Proposition 1** (Decrease of the residual variance)**.** *Suppose the conditions in Theorem 1 hold. For any iteration $t = 0, 1, \cdots$, we have*

$$2d\rho_t^2 \le \frac{1}{n} \sum_{i=1}^n \mathbb{E}\big[\big\|\boldsymbol{x}_i^{(t+\frac{1}{2})} - \boldsymbol{y}_i^{(t-1)}\big\|^2\big]$$

$$\le 2\left(1 + \sqrt{\frac{C}{\omega}} \frac{1 - \gamma_{t-1}\lambda_n}{\omega}\right)^2 (2G^2 + 2\sigma^2 + \mu^2 d)\eta_{t-1}^2 = \mathcal{O}\left(\frac{1}{t^{2\epsilon}}\right), \tag{31}$$

*where $C < 116$ is the constant defined in Theorem 1.*

*Proof.* See Appendix A.6.1. □

It follows from Proposition 1 that $\rho_t$ decreases with the order of $\mathcal{O}\left(\frac{1}{\sqrt{d}t^\epsilon}\right)$. Plugging this result into (31), the number of communication bits for the Laplace distributed residuals decreases to $\mathcal{O}\left(de^{-1/\Gamma}\right)$ with the rate of $\mathcal{O}(-d\log t)$ with high probability.

**Comparision with other compression methods**. ßwe compare the *worst-case* communication bit rate of our compression scheme with the existing method QSGD (Alistarh et al., 2017) and signSGD (Bernstein et al., 2018) in Table 1. Here, the worst-case communication complexity of QSGD follows from (Alistarh et al., 2017, Lemma A.4) by noting that every quantization output $\boldsymbol{q}_i^{(t)}$ satisfies $\|\boldsymbol{q}_i^{(t)}\|_0 \le d$.

Table 1 shows that our method attains a smaller communication cost than QSGD. This result algins with our numerical findings in Section 6, where our approach consistently outperforms QSGD in communication efficiency. On the other hand, when comparing our method with signSGD, with a small quantization precision $\Gamma$, our method exhibits comparable communication costs. Nevertheless, signSGD, limited to one-bit quantization only, incurs substantial quantization errors and subsequently slower training convergence. As demonstrated in Section 6, our method not only maintains similar communication costs but also significantly enhances training performance compared to signSGD.

A.4   PROOF OF THEOREM 1

Before proving Theorem 1, we state the following two lemmas.

**Lemma 3.** *Suppose $\gamma_t \leq \frac{1-|\lambda_2|}{4\tau}$, then for any $\theta > 0$ we have:*

$$\mathbb{E}_{Q_{t+1}} \left[ \left(1 + \theta^{-1}\right) \left\| \boldsymbol{Z}^{(t+1)} - \overline{\boldsymbol{Z}}^{(t+1)} \right\|_F^2 + \left\| \boldsymbol{X}^{(t+3/2)} - \boldsymbol{Y}^{(t+1)} \right\|_F^2 \right]$$

$$\leq (1 + \theta^{-1})(1 - \omega) \left( \left\| \boldsymbol{X}^{(t+1/2)} - \overline{\boldsymbol{X}}^{(t+1/2)} \right\|_F^2 + \left\| \boldsymbol{X}^{(t+1/2)} - \boldsymbol{Y}^{(t)} \right\|_F^2 \right)$$

$$+ (1+\theta)\eta_t^2 \left\| \nabla \boldsymbol{F} \left( \boldsymbol{X}^{(t+1)}, \xi^{(t+1)} + \boldsymbol{\Phi}^{(t+1)} \right) \right\|_F^2, \tag{32}$$

*where $\omega = \frac{(1-|\lambda_2|)^2}{8\tau} \leq \frac{1}{8}$.*

*Proof.* See Appendix A.6.2.   □

Lemma 3 provides a useful bound on the aggregation error in terms of the model averaging error and the quantization error.

**Lemma 4.** *Consider a sequence $\{r_t\}_{t\geq 1}$ s.t.*

$$r_{t+1} \leq (1 - c/2)r_t + \frac{2}{c}\eta_t^2 A, \tag{33}$$

*for positive constants $c$ and $A$; and $\eta_t = \frac{b}{(t+a)^\epsilon}$ with $a, b > 0$ and $\epsilon \in (0, 1]$.*

*Moreover, suppose $\epsilon < \frac{c}{4(2-c)}a$ and $r_0 \leq \frac{A\eta_0^2}{c} \frac{4(a+2\epsilon)}{ac+4(c-2)\epsilon}$. We have*

$$r_t \leq \frac{\eta_t^2 A}{c} \frac{4(a+2\epsilon)}{ac+4(c-2)\epsilon}, \forall t. \tag{34}$$

*Proof.* See Appendix A.6.3.   □

Equipped with the above results, we are ready to prove Theorem 1. Define the following auxiliary functions:

$$\mathcal{L}_{\eta_t,\gamma_t}(\boldsymbol{X}) = \sum_{i=1}^n \left( F_i(\boldsymbol{x}_i) + \frac{\gamma_t}{2\eta_t} \|\boldsymbol{x}_i\|_{\mathbf{I}-\boldsymbol{W}}^2 \right) = \mathbf{1}^T \boldsymbol{F}(\boldsymbol{X}) + \frac{\gamma_t}{2\eta_t} \|\boldsymbol{X}\|_{\mathbf{I}-\boldsymbol{W}}^2, \tag{35}$$

$$\mathcal{M}_{\eta_t,\gamma_t}(\boldsymbol{X}) = \mathcal{L}_{\eta_t,\gamma_t}(\boldsymbol{X}) + \mu \|\boldsymbol{X}\|_{1,1}. \tag{36}$$

Then, Step 9 of Alg. 2 can be represented as

$$\boldsymbol{X}^{(t+1)} = \text{prox}\left( (1 - \gamma_t)\boldsymbol{X}^{(t)} + \gamma_t \boldsymbol{Y}^{(t)}\boldsymbol{W} - \eta_t \nabla \boldsymbol{F}(\boldsymbol{X}^{(t)}, \xi^{(t)}) \right)$$

$$= \boldsymbol{X}^{(t)} - \eta_t \left( \underbrace{\nabla \boldsymbol{F}(\boldsymbol{X}^{(t)}, \xi^{(t)}) + \frac{\gamma_t}{\eta_t}\boldsymbol{X}^{(t)}(\mathbf{I} - \boldsymbol{W})}_{\nabla \mathcal{L}_{\eta_t,\gamma_t}(\boldsymbol{X}^{(t)};\xi_t)} + \boldsymbol{\Phi}^{(t+1)} \right) - \gamma_t \left( \underbrace{\boldsymbol{X}^{(t)} - \boldsymbol{Y}^{(t-1)} - Q(\boldsymbol{X}^{(t)} - \boldsymbol{Y}^{(t-1)})}_{\text{Quantization error} \triangleq \boldsymbol{E}_t} \right) \boldsymbol{W},$$

$$\tag{37}$$

for some subgradient $\boldsymbol{\Phi}^{(t+1)} \in \partial \left( \mu \|\boldsymbol{X}^{(t+1)}\|_{1,1} \right)$. We observe the following properties:

**Observation 1.** *Every subgradient of the regularization function $\phi \in \partial(\mu \|\boldsymbol{x}\|_1)$ has a bounded norm as $\|\phi\|_2 \leq \mu\sqrt{d}$. Under Assumption 3, we have $\mathbb{E}[\|\nabla F(\boldsymbol{X};\xi) + \boldsymbol{\Phi}\|_F^2] \leq 2n(2G^2 + 2\sigma^2 + \mu^2 d)$ for any $\phi_i \in \partial(\mu \|\boldsymbol{x}_i\|_1)$, $\boldsymbol{X}$, and $\xi$.*

**Observation 2.** *Under Assumption 3, $\mathcal{L}_{\eta_t,\gamma_t}(\cdot)$ is Lipschitz smooth with constant $L' = (1 - \gamma_t)L + \gamma_t\eta_t^{-1}(1 - \lambda_n)$.*

Define a sequence $\{r_t\}_t$ such that

$$r_t = \mathbb{E}\left[\left\|\boldsymbol{X}^{(t)} - \overline{\boldsymbol{X}}^{(t)}\right\|_F^2 + \left\|\boldsymbol{X}^{(t+1/2)} - \boldsymbol{Y}^{(t)}\right\|_F^2\right]. \tag{38}$$

At iteration $t+1$, we have

$$r_{t+1} = \mathbb{E}\left[\left\|\boldsymbol{Z}^{(t+1)} - \overline{\boldsymbol{Z}}^{(t+1)} + \eta_{t+1}\left(\Phi^{(t+1)}\right)\left(\frac{\mathbf{1}\mathbf{1}^T}{n} - \mathbf{I}\right)\right\|_F^2\right] + \mathbb{E}\left[\left\|\boldsymbol{X}^{(t+3/2)} - \boldsymbol{Y}^{(t+1)}\right\|_F^2\right]$$

$$\leq \mathbb{E}\left[(1+\alpha^{-1})\left\|\boldsymbol{Z}^{(t+1)} - \overline{\boldsymbol{Z}}^{(t+1)}\right\|_F^2 + \left\|\nabla\boldsymbol{X}^{(t+1)} - \boldsymbol{Y}^{(t+1)}\right\|_F^2\right] + (1+\alpha)\eta_t^2\mathbb{E}\left[\left\|\Phi^{(t+1)}\right\|_F^2\right]\left(\left\|\frac{\mathbf{1}\mathbf{1}^T}{n} - \mathbf{I}\right\|_2^2\right)$$

$$\overset{(32),\theta=\alpha}{\leq} (1+\alpha^{-1})(1-\omega)\left\|\boldsymbol{X}^{(t+1/2)} - \overline{\boldsymbol{X}}^{(t+1/2)}\right\|_F^2 + (1-\omega)\left\|\boldsymbol{X}^{(t+1/2)} - \boldsymbol{Y}^{(t)}\right\|_F^2$$

$$+ (1+\alpha)(\eta_t^2)2n\mu^2 d + (1+\alpha)\eta_t^2\left\|\nabla F\left(\boldsymbol{X}^{(t+1)}, \xi^{(t+1)}\right) + \Phi^{(t+1)}\right\|_F^2$$

$$\leq (1+\alpha^{-1})(1+\beta^{-1})(1-\omega)\left\|\boldsymbol{X}^{(t)} - \overline{\boldsymbol{X}}^{(t)}\right\|_F^2 + (1+\beta)(1-\omega)\eta_t^2\left\|F(\boldsymbol{X}^{(t+1)})\right\|_F^2$$

$$+ (1+\alpha)(\eta_t^2)2n\mu^2 d + (1-\omega)\left\|\boldsymbol{X}^{(t+1/2)} - \boldsymbol{Y}^{(t)}\right\|_F^2 + (1+\alpha)\eta_t^2\left\|\nabla F\left(\boldsymbol{X}^{(t+1)}, \xi^{(t+1)}\right) + \Phi^{(t+1)}\right\|_F^2$$

$$\overset{\beta=6/(\omega(1-\omega))}{\leq} (1+\alpha^{-1})(1 + \frac{\omega(1-\omega)}{6})(1-\omega)r_t + (1+\alpha)(1 + \frac{6}{\omega(1-\omega)})(1-\omega)4n\eta_t^2(2G^2 + 2\sigma^2 + \mu^2 d)$$

$$\overset{\alpha=\omega/(3-\omega)}{\leq} (1+\omega/2)(1-\omega)r_t + \left(\frac{9}{\omega}\right)4n\eta_t^2(2G^2 + 2\sigma^2 + \mu^2 d)$$

$$\leq (1-\omega/2)r_t + \frac{36}{\omega}n\eta_t^2(2G^2 + 2\sigma^2 + \mu^2 d).$$

Combining the recursion of $r_{t+1}$ and Lemma 4, we have (33) with $A = 18n(2G^2 + 2\sigma^2 + \mu^2 d)$. Applying Lemma 4 and assuming $\epsilon < \frac{\omega}{4(2-\omega)}a$, we have

$$r_t \leq \frac{56(a+2\epsilon)}{\omega(a\omega + 4(\omega-2)\epsilon)}n\eta_t^2(2G^2 + 2\sigma^2 + \mu^2 d). \tag{39}$$

To prove (13), we first note that if $a \geq 8\epsilon/\omega$,

$$\frac{a}{\epsilon} \geq \frac{8}{\omega} > \frac{4(2-\omega)}{\omega}.$$

(39) follows from the assumption of $\left\|\boldsymbol{x}^{(0)}\right\| = 0$. Furthermore, define the function $h(x) = \frac{1+2x}{\omega+4(\omega-2)x}$. We have

$$h'(x) = 2 \cdot \frac{\omega + 4(\omega-2)x - 2(1+2x)(\omega-2)}{(\omega + 4(\omega-2)x)^2} = \frac{8-2\omega}{(\omega + 4(\omega-2)x)^2} > 0.$$

For $\epsilon/a \leq \omega/8$, we have

$$h(\epsilon/a) \leq h(\omega/8) = \frac{2+\omega/2}{\omega^2} \leq \frac{33}{16\omega^2}, \tag{40}$$

where the last inequality follows from $\omega \leq 1/8$. Substituting (40) into (39) and applying $\mathbb{E}[\left\|\boldsymbol{X}^t - \overline{\boldsymbol{X}}^t\right\|_F^2] \leq r_t$, we have

$$\mathbb{E}\left\|\boldsymbol{X}^{(t)} - \overline{\boldsymbol{X}}^{(t)}\right\|_F^2 \leq r_t \leq \frac{56}{\omega}n\eta_t^2(2G^2 + 2\sigma^2 + \mu^2 d)h(\omega/8) \leq \frac{C}{\omega^3}n\eta_t^2(2G^2 + 2\sigma^2 + \mu^2 d), \tag{41}$$

where $C < 116$ is a constant.

### A.5 PROOF OF THEOREM 2

Theorem 1 implies the following results that will be useful in the proof of Theorem 2.

**Proposition 2.** *Under the conditions of Theorem 1, for any $t = 0, 1, \cdots$, we have*

$$\frac{1}{n} \sum_{i=1}^{n} \mathbb{E}\left[\left\|\boldsymbol{x}_i^{(t+1)} - \boldsymbol{y}_i^{(t)}\right\|^2\right] \leq 2\left(1 + \sqrt{\frac{C}{\omega}}\frac{1 - \gamma_t \lambda_n}{\omega}\right)^2 (2G^2 + 2\sigma^2 + \mu^2 d)\eta_t^2, \quad (42)$$

*where $C < 116$ is the constant defined in Theorem 1.*

*Proof.* See Appendix A.6.4. □

**Proposition 3.** *Under the same conditions of Theorem 1, one has that*

$$\mathbb{E}\left\|\overline{\boldsymbol{x}}^{(t+1)} - \overline{\boldsymbol{x}}^{(t)}\right\|^2 \leq \frac{1}{n}\sum_{i=1}^{n} \mathbb{E}\left\|\boldsymbol{x}_i^{(t+1)} - \boldsymbol{x}_i^{(t)}\right\|^2 \leq (2G^2 + 2\sigma^2 + \mu^2 d)\left(1 + 6\sqrt{\frac{C}{32\omega^3} + 1}\right)^2 \eta_t^2,$$

$$(43)$$

*where $C < 116$ is the constant defined in Theorem 1.*

*Proof.* Note that

$$\left\|\overline{\boldsymbol{X}}^{(t+1)} - \overline{\boldsymbol{X}}^{(t)}\right\|_F^2 = \left\|(\boldsymbol{X}^{(t+1)} - \boldsymbol{X}^{(t)})\mathbf{1}\mathbf{1}^T/n\right\|_F^2 \leq \left\|\boldsymbol{X}^{(t+1)} - \boldsymbol{X}^{(t)}\right\|_F^2. \quad (44)$$

Applying (22), for any $\alpha, \beta > 0$,

$$\mathbb{E}\left\|\boldsymbol{X}^{(t+1)} - \boldsymbol{X}^{(t)}\right\|_F^2 \leq (1 + \alpha^{-1})\mathbb{E}\left\|\boldsymbol{X}^{(t+1)} - \overline{\boldsymbol{X}}^{(t+1)}\right\|_F^2$$

$$+ (1 + \alpha)(1 + \beta^{-1})\mathbb{E}\left\|\boldsymbol{X}^{(t)} - \overline{\boldsymbol{X}}^{(t)}\right\|_F^2 + (1 + \alpha)(1 + \beta)\mathbb{E}\left\|\overline{\boldsymbol{X}}^{(t+1)} - \overline{\boldsymbol{X}}^{(t)}\right\|_F^2.$$

$$(45)$$

The last term in (45) can be simplified as

$$\mathbb{E}\left\|\overline{\boldsymbol{X}}^{(t+1)} - \overline{\boldsymbol{X}}^{(t)}\right\|_F^2 \leq \mathbb{E}\left\|\eta_t\left(\nabla\boldsymbol{F}(\boldsymbol{X}^{(t)}, \xi^{(t)}) + \Phi^{(t+1)}\right) - \gamma_t(\boldsymbol{X}^{(t)} - \boldsymbol{Y}^{(t)})\right\|_F^2 \left\|\mathbf{1}\mathbf{1}^T/n\right\|_2^2$$

$$\leq 2\eta_t^2 \mathbb{E}\left\|\nabla\boldsymbol{F}(\boldsymbol{X}^{(t)}, \xi^{(t)}) + \Phi^{(t+1)}\right\|_F^2 + 2\gamma_t^2 \mathbb{E}\left\|\boldsymbol{X}^{(t)} - \boldsymbol{Y}^{(t)}\right\|_F^2$$

$$\leq 4n(2G^2 + 2\sigma^2 + \mu^2 d)\eta_t^2 + 2\gamma_t^2 \mathbb{E}\left\|\boldsymbol{X}^{(t)} - \boldsymbol{Y}^{(t)}\right\|_F^2. \quad (46)$$

Combining (45), (46), and (38), for $\forall \alpha, \beta > 0$,

$$\mathbb{E}[\left\|\boldsymbol{X}^{(t+1)} - \boldsymbol{X}^{(t)}\right\|_F^2] \leq (1 + \alpha^{-1})r_{t+1} + (1 + \alpha)r_t \max\{1 + \beta^{-1}, 2\gamma_t^2(1 + \beta)\}$$

$$+ (1 + \alpha)(1 + \beta)4n(2G^2 + 2\sigma^2 + \mu^2 d)\eta_t^2. \quad (47)$$

In particular, setting $\beta = 8$, we have $1 + \beta^{-1} = 9/8$ and $2\gamma_t^2(1 + \beta) = 18/\gamma_t^2 \leq 9/8$. Applying (41),

$$\mathbb{E}[\left\|\boldsymbol{X}^{(t+1)} - \boldsymbol{X}^{(t)}\right\|_F^2] \leq \left(1 + \alpha^{-1} + (1 + \alpha)\left(\frac{9C}{8\omega^3} + 36\right)\right)n(2G^2 + 2\sigma^2 + \mu^2 d)\eta_t^2. \quad (48)$$

By the inequality of arithmetic and geometric means, we have

$$1 + \alpha^{-1} + (1 + \alpha)\left(\frac{9C}{8\omega^3} + 36\right) \geq \left(1 + \sqrt{\frac{9C}{8\omega^3} + 36}\right)^2, \quad (49)$$

where the inequality holds if $\alpha^{-2} = \frac{9C}{8\omega^3} + 36$. Combining (48) and (49) completes the proof. □

We prove the convergence of MALCOM-PSGD by using Theorem 1, Proposition 2, and Proposition 3 as follows. First, note that $\eta_t \leq \frac{1}{a^\epsilon} \leq \frac{1 - (1 - \lambda_n)\gamma_t}{L}, \forall t$ with $\gamma_t \leq \frac{1 - a^{-\epsilon}}{1 - \lambda_n}$. Leveraging $L' = L + \gamma_t \eta_t^{-1}(1 - \lambda_n)$ in Observation 2, we have $L' \leq \eta_t^{-1}$. Following (Zeng & Yin, 2018a, Eqs. (88)-(89)), for any given $\boldsymbol{U} \in \mathbb{R}^{d \times n}$ that is independent to $t$,

$$\mathcal{M}_{\eta_t, \gamma_t}(\boldsymbol{U}) - \mathcal{M}_{\eta_t, \gamma_t}(\boldsymbol{X}^{(t+1)}) \geq \left\langle \nabla\mathcal{L}_{\eta_t, \gamma_t}(\boldsymbol{X}^{(t)}) + \Phi^{(t+1)}, \boldsymbol{U} - \boldsymbol{X}^{(t+1)}\right\rangle - \frac{L'}{2}\left\|\boldsymbol{X}^{(t+1)} - \boldsymbol{X}^{(t)}\right\|_F^2,$$

where $\boldsymbol{\Phi}^{(t+1)} \in \partial(\mu \left\| \boldsymbol{X}^{(t+1)} \right\|_{1,1})$. Substituting $\boldsymbol{\Phi}^{(t+1)}$ in (37) and taking $\boldsymbol{U}$ to be an optimal solution $\boldsymbol{X}^{\star} \in \mathcal{X}^{\star}$,

$$\mathbb{E}[\mathcal{M}_{\eta_t,\gamma_t}(\boldsymbol{X}^{(t+1)}) - \mathcal{M}_{\eta_t,\gamma_t}(\boldsymbol{X}^{\star})]$$

$$\leq -\eta_t^{-1} \mathbb{E}\left\langle \boldsymbol{X}^{(t)} - \boldsymbol{X}^{(t+1)} - \gamma_t(\boldsymbol{X}^{(t)} - \boldsymbol{Y}^{(t)})\boldsymbol{W}, \boldsymbol{X}^{\star} - \boldsymbol{X}^{(t+1)} \right\rangle + \frac{L'}{2}\mathbb{E}\left\| \boldsymbol{X}^{(t+1)} - \boldsymbol{X}^{(t)} \right\|_F^2$$

$$\underbrace{-\left\langle \mathbb{E}_t[\nabla\mathcal{L}_{\eta_t,\gamma_t}(\boldsymbol{X}^{(t)}) - \nabla\mathcal{L}_{\eta_t,\gamma_t}(\boldsymbol{X}^{(t)};\xi^t)], \boldsymbol{X}^{\star} - \mathbb{E}_{t+1}[\boldsymbol{X}^{(t+1)}] \right\rangle}_{=0}$$

$$\overset{L' \leq \eta_t^{-1}}{\leq} \frac{\eta_t^{-1}}{2}\mathbb{E}\left\| \boldsymbol{X}^{(t+1)} - \boldsymbol{X}^{(t)} \right\|_F^2 - \eta_t^{-1}\mathbb{E}\left\langle \boldsymbol{X}^{(t)} - \boldsymbol{X}^{(t+1)} - \gamma_t(\boldsymbol{X}^{(t)} - \boldsymbol{Y}^{(t)})\boldsymbol{W}, \boldsymbol{X}^{\star} - \boldsymbol{X}^{(t+1)} \right\rangle$$

$$= \frac{1}{2\eta_t}(\mathbb{E}\left\| \boldsymbol{X}^{(t)} - \boldsymbol{X}^{\star} \right\|_F^2 - \mathbb{E}\left\| \boldsymbol{X}^{(t+1)} - \boldsymbol{X}^{\star} \right\|_F^2) + \gamma_t\eta_t^{-1}\mathbb{E}\left\langle (\boldsymbol{X}^{(t)} - \boldsymbol{Y}^{(t)})\boldsymbol{W}, \boldsymbol{X}^{\star} - \boldsymbol{X}^{(t+1)} \right\rangle$$

$$\leq \frac{1}{2\eta_t}\left( \mathbb{E}\left\| \boldsymbol{X}^{(t)} - \boldsymbol{X}^{\star} \right\|_F^2 - \mathbb{E}\left\| \boldsymbol{X}^{(t+1)} - \boldsymbol{X}^{\star} \right\|_F^2 + 2\gamma_t\mathbb{E}_t\left\| \boldsymbol{X}^{(t)} - \boldsymbol{Y}^{(t)} \right\|_F \mathbb{E}_{t+1}\left\| \boldsymbol{X}^{\star} - \boldsymbol{X}^{(t+1)} \right\|_F \right)$$

$$\overset{\text{Pro. 2}}{\leq} \frac{1}{2\eta_t}\left( \mathbb{E}\left\| \boldsymbol{X}^{(t)} - \boldsymbol{X}^{\star} \right\|_F^2 - \mathbb{E}\left\| \boldsymbol{X}^{(t+1)} - \boldsymbol{X}^{\star} \right\|_F^2 + 2C_0 \underbrace{\gamma_t\eta_t\mathbb{E}_{t+1}\left\| \boldsymbol{X}^{\star} - \boldsymbol{X}^{(t+1)} \right\|_F}_{\triangleq A_{t+1}} \right),$$

$$(50)$$

where $C_0$ is a constant given by the square root of the r.h.s. of Theorem 1.

Taking the summation of (50) recursively w.r.t. $k = 0, 1, ..., t$ and using $\boldsymbol{X}^{(0)} = \boldsymbol{0}$, we have

$$\sum_{k=0}^{t} \eta_k(\mathbb{E}[\mathcal{M}_{\eta_k,\gamma_k}(\boldsymbol{X}^{(k+1)})] - n\mathcal{F}^{\star})) \leq \frac{1}{2}\left\| \boldsymbol{X}^{\star} \right\|_F^2 + C_0\sum_{k=0}^{t} A_{k+1}. \tag{51}$$

On the other hand, note that

$$\left\| \boldsymbol{X}^{(k+1)} \right\|_{1,1} \geq \left\| \overline{\boldsymbol{X}}^{(k+1)} \right\|_{1,1} - \left\| \boldsymbol{X}^{(k+1)} - \overline{\boldsymbol{X}}^{(k+1)} \right\|_{1,1} \geq \left\| \overline{\boldsymbol{X}}^{(k+1)} \right\|_{1,1} - \sqrt{d}\left\| \boldsymbol{X}^{(k+1)} - \overline{\boldsymbol{X}}^{(k+1)} \right\|_F; \tag{52}$$

and

$$\mathcal{L}_{\eta_k,\gamma_k}(\boldsymbol{X}^{(k+1)}) \geq \mathcal{L}_{\eta_k,\gamma_k}(\overline{\boldsymbol{X}}^{(k+1)}) + <\nabla\mathcal{L}_{\eta_k,\gamma_k}(\boldsymbol{X}^{(k+1)}), \boldsymbol{X}^{(k+1)} - \overline{\boldsymbol{X}}^{(k+1)}> -\frac{L'}{2}\left\| \boldsymbol{X}^{(k+1)} - \overline{\boldsymbol{X}}^{(k+1)} \right\|_F^2$$

$$\geq \mathcal{L}_{\eta_k,\gamma_k}(\overline{\boldsymbol{X}}^{(k+1)}) - G\left\| \boldsymbol{X}^{(k+1)} - \overline{\boldsymbol{X}}^{(k+1)} \right\|_F - \frac{L'}{2}\left\| \boldsymbol{X}^{(k+1)} - \overline{\boldsymbol{X}}^{(k+1)} \right\|_F^2. \tag{53}$$

Combining (52) and (53) and noting that $\overline{\boldsymbol{X}}$ is consensus by definition, we have

$$\mathcal{M}_{\eta_k,\gamma_k}(\boldsymbol{X}^{(k+1)}) \geq n\mathcal{F}(\overline{\boldsymbol{X}}^{(k+1)}) - (G + \mu\sqrt{d})\left\| \boldsymbol{X}^{(k+1)} - \overline{\boldsymbol{X}}^{(k+1)} \right\|_F - \frac{L'}{2}\left\| \boldsymbol{X}^{(k+1)} - \overline{\boldsymbol{X}}^{(k+1)} \right\|_F^2. \tag{54}$$

Combining (51) and (54) and applying $L' \leq \eta_t^{-1}$,

$$n\sum_k \eta_k(\mathbb{E}\mathcal{F}(\overline{\boldsymbol{X}}^{(k+1)}) - \mathcal{F}^{\star}) \leq \frac{1}{2}\left\| \boldsymbol{X}^{\star} \right\|_F^2 + (G + \mu\sqrt{d})\sum_k \eta_k\mathbb{E}\left\| \boldsymbol{X}^{(k+1)} - \overline{\boldsymbol{X}}^{(k+1)} \right\|_F$$

$$+ \frac{1}{2}\sum_k \mathbb{E}\left\| \boldsymbol{X}^{(k+1)} - \overline{\boldsymbol{X}}^{(k+1)} \right\|_F^2 + C_0\sum_{k=0}^{t} A_{k+1}. \tag{55}$$

Applying Theorem 1 and Proposition 3, we have $\mathbb{E}\left\| \boldsymbol{X}^{(k+1)} - \overline{\boldsymbol{X}}^{(k+1)} \right\|_F = \mathcal{O}(\eta_k)$ and $\mathbb{E}\left\| \boldsymbol{X}^{(k+1)} - \overline{\boldsymbol{X}}^{(k+1)} \right\|_F^2 = \mathcal{O}(\eta_k^2)$. Note that the left-hand side of (55) equals to $n(\sum_k \eta_k)(\overline{\mathcal{F}}_t -$

$\mathcal{F}^\star$). Therefore,

$$
\mathbb{E}\overline{\mathcal{F}}_t - \mathcal{F}^\star \leq \frac{\frac{n}{2}\left\|\boldsymbol{x}^\star\right\|_2^2 + nC_1'\sum_{k=0}^t \eta_k^2 + \sqrt{n}C_2'\sum_{k=0}^t \gamma_k\eta_k\mathbb{E}\left\|\boldsymbol{X}^\star - \boldsymbol{X}^{(k+1)}\right\|_F}{n(\sum_{k=0}^t \eta_k)}
$$

$$
= \frac{C_3}{\sum_{k=0}^t \eta_k} + C_1'\frac{\sum_{k=0}^t \eta_k^2}{\sum_{k=0}^t \eta_k} + C_2'\frac{\sum_{k=0}^t \gamma_k\eta_k\mathbb{E}\left\|\boldsymbol{X}^\star - \boldsymbol{X}^{(k+1)}\right\|_F}{\sqrt{n}(\sum_{k=0}^t \eta_k)}, \tag{56}
$$

where

$$
C_1' \leq \frac{C}{2\omega^3}\left(2G^2 + 2\sigma^2 + \mu^2 d\right),
$$

$$
C_2' \leq \sqrt{\frac{C}{2\omega^3}\left(2G^2 + 2\sigma^2 + \mu^2 d\right)},
$$

$$
C_3 = \left\|\boldsymbol{x}^\star\right\|^2/2,
$$

with $C < 116$ be the constant defined in Theorem 1. The final step to prove (15) is to compute the last term in (56). Applying the triangle inequality,

$$
\gamma_k\eta_k\mathbb{E}\left\|\boldsymbol{X}^\star - \boldsymbol{X}^{(k+1)}\right\|_F \leq (\left\|\boldsymbol{X}^\star\right\|_F + \mathbb{E}\left\|\boldsymbol{X}^{(k+1)}\right\|_F)\gamma_k\eta_k \leq (B + \left\|\boldsymbol{X}^\star\right\|_F)\eta_k^2
$$

$$
= B\eta_k^2 + \sqrt{n}\left\|\boldsymbol{x}^\star\right\|_2\eta_k^2, \tag{57}
$$

where $B$ is defined in Assumption 4, and the last inequality follows from $\gamma_t \leq \eta_t$. Plugging this result into (56), we have

$$
\mathbb{E}\overline{\mathcal{F}}_t - \mathcal{F}^\star \leq \frac{C_3}{\sum_{k=0}^t \eta_k} + (C_1' + \left\|\boldsymbol{x}^\star\right\|_2 C_2')\frac{\sum_{k=0}^t \eta_k^2}{\sum_{k=0}^t \eta_k} + \frac{C_2'B}{\sqrt{n}}\frac{\sum_{k=0}^t \eta_k^2}{\sum_{k=0}^t \eta_k}, \tag{58}
$$

which is identical to (15). Finally, applying the result in (Chen, 2012, Sect. 3.2.4) to characterize the order of $\frac{1}{\sum_{k=1}^\infty \eta_k}$ and $\frac{\sum_{k=1}^t \eta_k^2}{\sum_{k=1}^t \eta_k}$, the terms in (58) exhibit the order of

$$
\frac{1}{\sum_{k=1}^t \eta_k} = \begin{cases} \mathcal{O}(\frac{1}{t^{1-\epsilon}}) & \text{if } \epsilon \in (0, \frac{1}{2}), \\ \mathcal{O}(\frac{1}{\sqrt{t}}) & \text{if } \epsilon = \frac{1}{2}, \\ \mathcal{O}(\frac{1}{t^{(1-\epsilon)}}) & \text{if } \epsilon \in (\frac{1}{2}, 1), \\ \mathcal{O}(\frac{1}{\ln t}) & \text{if } \epsilon = 1. \end{cases}, \quad \frac{\sum_{k=1}^t \eta_k^2}{\sum_{k=1}^t \eta_k} = \begin{cases} \mathcal{O}(\frac{1}{t^\epsilon}) & \text{if } \epsilon \in (0, \frac{1}{2}), \\ \mathcal{O}(\frac{\ln t}{\sqrt{t}}) & \text{if } \epsilon = \frac{1}{2}, \\ \mathcal{O}(\frac{1}{t^{1-\epsilon}}) & \text{if } \epsilon \in (\frac{1}{2}, 1), \\ \mathcal{O}(\frac{1}{\ln t}) & \text{if } \epsilon = 1. \end{cases}
$$

This completes the proof.

## A.6  Lemma and Proposition Proofs

### A.6.1  Proof of Proposition 1

When the entries $\boldsymbol{x}_i^{(t+\frac{1}{2})} - \boldsymbol{y}_i^{(t-1)}$ follow the i.i.d. Laplace distribution with the diversity parameter $\rho_t$, the variance of each entry is given by $2\rho_t^2$. Therefore, we have

$$
2d\rho_t^2 \leq \frac{1}{n}\sum_{i=1}^n \mathbb{E}[\left\|\boldsymbol{x}_i^{(t+\frac{1}{2})} - \boldsymbol{y}_i^{(t-1)}\right\|^2] = \mathbb{E}\left[\left\|\boldsymbol{X}^{(t+\frac{1}{2})} - \boldsymbol{Y}^{(t-1)}\right\|_F^2\right]. \tag{59}
$$

For ease of notation, we bound it with iteration $t + 1$. For any $\alpha > 0$, we have

$$\mathbb{E}\left[\left\|\boldsymbol{X}^{(t+1+\frac{1}{2})} - \boldsymbol{Y}^{(t)}\right\|_F^2\right] = \mathbb{E}\left[\left\|\boldsymbol{Z}^{(t+1)} - \boldsymbol{Y}^{(t)} - \eta_t\left(\nabla \boldsymbol{F}(\boldsymbol{X}^{(t+1)}; \xi_t) + \Phi^{(t+1)}\right)\right\|_F^2\right]$$

$$\leq (1 + \alpha^{-1})\eta_t^2 \mathbb{E}\left[\left\|\nabla \boldsymbol{F}(\boldsymbol{X}^{(t+1)}; \xi_t) + \Phi^{(t+1)}\right\|_F^2\right] + (1 + \alpha)\mathbb{E}\left[\left\|(1 - \gamma_t)\boldsymbol{X}^{(t)} - \boldsymbol{Y}^{(t)} + \gamma_t \boldsymbol{Y}^{(t)}\boldsymbol{W}\right\|_F^2\right]$$

$$\leq 2(1 + \alpha^{-1})(2G^2 + 2\sigma^2 + \mu^2 d)n\eta_t^2 + (1 + \alpha)\mathbb{E}\left[\left\|(\boldsymbol{I} - \gamma_t \boldsymbol{W})(\boldsymbol{X}^{(t)} - \boldsymbol{Y}^{(t)}) + \gamma_t \boldsymbol{X}^{(t)}(\boldsymbol{I} - \boldsymbol{W})\right\|_F^2\right]$$

$$\overset{(a)}{\leq} 2(1 + \alpha)\left(\mathbb{E}\left[\left\|\boldsymbol{X}^{(t)} - \boldsymbol{Y}^{(t)}\right\|_F^2\right]\|\boldsymbol{I} - \gamma_t \boldsymbol{W}\|_2^2 + \gamma_t^2 \mathbb{E}\left[\left\|(\boldsymbol{X}^{(t)} - \overline{\boldsymbol{X}}^{(t)})(\boldsymbol{I} - \boldsymbol{W})\right\|_F^2\right]\right)$$
$$+ 2(1 + \alpha^{-1})(2G^2 + 2\sigma^2 + \mu^2 d)n\eta_t^2$$

$$\leq 2(1 + \alpha)\left((1 - \gamma_t\lambda_n)^2 \mathbb{E}\left[\left\|\boldsymbol{X}^{(t)} - \boldsymbol{Y}^{(t)}\right\|_F^2\right] + \gamma_t^2(1 - \lambda_n)^2 \mathbb{E}\left[\left\|\boldsymbol{X}^{(t)} - \overline{\boldsymbol{X}}^{(t)}\right\|_F^2\right]\right)$$
$$+ 2(1 + \alpha^{-1})(2G^2 + 2\sigma^2 + \mu^2 d)n\eta_t^2$$

$$\overset{(b)}{\leq} 2(1 + \alpha)(1 - \gamma_t\lambda_n)^2 \mathbb{E}\left[\left\|\boldsymbol{X}^{(t)} - \overline{\boldsymbol{X}}^{(t)}\right\|_F^2 + \left\|\boldsymbol{X}^{(t)} - \boldsymbol{Y}^{(t)}\right\|_F^2\right] + 2(1 + \alpha^{-1})(2G^2 + 2\sigma^2 + \mu^2 d)n\eta_t^2$$

$$\overset{(c)}{\leq} 2(1 + \alpha)(1 - \gamma_t\lambda_n)^2 r_t + 2(1 + \alpha^{-1})(2G^2 + 2\sigma^2 + \mu^2 d)n\eta_t^2$$

$$\overset{(d)}{\leq} 2\left(\underbrace{1 + \alpha^{-1} + (1 + \alpha)(1 - \gamma_t\lambda_n)^2 C/\omega^3}_{\triangleq h(\alpha)}\right)(2G^2 + 2\sigma^2 + \mu^2 d)n\eta_t^2, \tag{60}$$

where $(a)$ follows from $\overline{\boldsymbol{X}}^{(t)}(\boldsymbol{I} - \boldsymbol{W}) = 0$; $(b)$ follows from $\gamma_t(1 - \lambda_n) \leq 1 - \gamma_t\lambda_n$; $(c)$ follows from the definition in (38); and $(d)$ follows from (41).

By the inequality of arithmetic and geometric means,

$$h(\alpha) = 1 + (1 - \gamma_t\lambda_n)^2 C/\omega^3 + \alpha^{-1} + \alpha(1 - \gamma_t\lambda_n)^2 C/\omega^3$$
$$\geq 1 + (1 - \gamma_t\lambda_n)^2 C/\omega^3 + 2(1 - \gamma_t\lambda_n)\sqrt{C/\omega^3}$$
$$= (1 + (1 - \gamma_t\lambda_n)\sqrt{C/\omega^3})^2, \tag{61}$$

where the inequality holds if $\alpha = \sqrt{\omega^3/C}/(1 - \gamma_t\lambda_n)$. Combining (60) and (61) completes the proof.

### A.6.2 PROOF OF LEMMA 3

For any $\alpha_1 > 0$, we have

$$\left\|\boldsymbol{Z}^{(t+1)} - \overline{\boldsymbol{Z}}^{(t+1)}\right\|_F^2$$

$$= \left\|(1 - \gamma_t)(\boldsymbol{X}^{(t+1/2)} - \overline{\boldsymbol{X}}^{(t+1/2)}) + \gamma_t(\boldsymbol{Y}^{(t)} - \boldsymbol{X}^{(t+1/2)})(\boldsymbol{W} - \boldsymbol{1}\boldsymbol{1}^T/n) + \gamma_t(\boldsymbol{X}^{(t+1)} - \overline{\boldsymbol{X}}^{(t+1/2)})(\boldsymbol{W} - \boldsymbol{1}\boldsymbol{1}^T/n)\right\|_F^2$$

$$\overset{(22)}{\leq} (1 + \alpha_1)\left\|(\boldsymbol{X}^{(t+1/2)} - \overline{\boldsymbol{X}}^{(t+1/2)})((1 - \gamma_t)\boldsymbol{I} + \gamma_t(\boldsymbol{W} - \boldsymbol{1}\boldsymbol{1}^T/n))\right\|_F^2 + (1 + \alpha_1^{-1})\gamma_t^2\lambda_2^2\left\|\boldsymbol{Y}^{(t)} - \boldsymbol{X}^{(t+1)}\right\|_F^2$$

$$\overset{(25),(23)}{=} (1 + \alpha_1)\left\|(\boldsymbol{X}^{(t+1/2)} - \overline{\boldsymbol{X}}^{(t+1/2)})((1 - \gamma_t)\boldsymbol{I} + \gamma_t(\boldsymbol{W} - \boldsymbol{1}\boldsymbol{1}^T/n))\right\|_F^2 + (1 + \alpha_1^{-1})\gamma_t^2\lambda_2^2\left\|\boldsymbol{Y}^{(t)} - \boldsymbol{X}^{(t+1/2)}\right\|_F^2. \tag{62}$$

Applying Jensen's inequality, we can bound the first term on the right-hand side above as

$$\left\| \left( \boldsymbol{X}^{(t+1/2)} - \overline{\boldsymbol{X}}^{(t+1/2)} \right) \left( (1 - \gamma_t)\,\mathbf{I} + \gamma_t \left( \boldsymbol{W} - \frac{1}{n}\mathbf{1}\mathbf{1}^\top \right) \right) \right\|_F$$

$$\overset{(24)}{\leq} (1 - \gamma_t) \left\| \boldsymbol{X}^{(t+1/2)} - \overline{\boldsymbol{X}}^{(t+1/2)} \right\|_F + \gamma_t |\lambda_2| \left\| \boldsymbol{X}^{(t+1/2)} - \overline{\boldsymbol{X}}^{(t)} \right\|_F$$

$$\leq (1 - \gamma_t(1 - |\lambda_2|)) \left\| \boldsymbol{X}^{(t+1/2)} - \overline{\boldsymbol{X}}^{(t+1/2)} \right\|_F. \tag{63}$$

Denoting $\tilde{\lambda} = 1 - |\lambda_2|$ and substituting 63 into equation 62, we have

$$\left\| \boldsymbol{Z}^{(t+1)} - \overline{\boldsymbol{Z}}^{(t+1)} \right\|_F^2 \leq (1 + \alpha_1)\left(1 - \gamma_t\tilde{\lambda}\right)^2 \left\| \boldsymbol{X}^{(t+1/2)} - \overline{\boldsymbol{X}}^{(t+1/2)} \right\|_F^2$$

$$+ \left(1 + \alpha_1^{-1}\right)\gamma_t^2\left(1 - \tilde{\lambda}\right)^2 \left\| \boldsymbol{Y}^{(t)} - \boldsymbol{X}^{(t+1/2)} \right\|_F^2. \tag{64}$$

On the other hand, for any $\alpha_2, \theta > 0$,

$$\mathbb{E}_Q\left[\left\| \boldsymbol{X}^{(t+3/2)} - \boldsymbol{Y}^{(t+1)} \right\|_F^2\right] = \mathbb{E}_Q\left[\left\| \boldsymbol{X}^{(t+3/2)} - Q\left( \boldsymbol{X}^{(t+3/2)} - \boldsymbol{Y}^{(t)} \right) - \boldsymbol{Y}^{(t)} \right\|_F^2\right]$$

$$\overset{\text{Lemma 2}}{\leq} (1 - \tau^{-1}) \left\| \boldsymbol{X}^{(t+3/2)} - \boldsymbol{Y}^{(t)} \right\|_F^2$$

$$= (1 - \tau^{-1}) \left\| \mathrm{prox}_{\eta_t, \mu\|\cdot\|_1}\left( (1 - \gamma_t)\boldsymbol{X}^{(t+1/2)} + \gamma_t\boldsymbol{Y}^{(t)}\boldsymbol{W} \right) - \boldsymbol{Y}^{(t)} - \eta_{t+1}\nabla\boldsymbol{F}(\boldsymbol{X}^{(t+1)}, \zeta^{(t+1)}) \right\|_F^2$$

$$= (1 + \theta^{-1})(1 - \tau^{-1}) \left\| (1 - \gamma_t)\boldsymbol{X}^{(t+1/2)} + \gamma_t\boldsymbol{Y}^{(t)}\boldsymbol{W} - \boldsymbol{Y}^{(t)} \right\|_F^2 \tag{65}$$

$$+ (1 + \theta)(1 - \tau^{-1})\eta_{t+1}^2 \left\| \nabla\boldsymbol{F}(\boldsymbol{X}^{(t+1)}, \xi^{(t+1)}) + \boldsymbol{\Phi}^{(t+1)} \right\|_F^2$$

$$\overset{\tau \geq 0}{\leq} \underbrace{(1 + \theta^{-1})\left(1 - \tau^{-1}\right) \left\| (1 - \gamma_t)\boldsymbol{X}^{(t+1/2)} + \gamma_t\boldsymbol{Y}^{(t)}\boldsymbol{W} - \boldsymbol{Y}^{(t)} \right\|_F^2}_{\triangleq D} + (1 + \theta)\eta_t^2 \left\| \nabla\boldsymbol{F}(\boldsymbol{X}^{(t+1)}, \xi^{(t+1)}) + \boldsymbol{\Phi}^{(t+1)} \right\|_F^2.$$

$$\tag{66}$$

Recall that where $\boldsymbol{\Phi}^{(t+1)} \in \partial(\mu\left\| \boldsymbol{X}^{(t+1)} \right\|_{1,1})$ Here, the term $D$ can be simplified as

$$D = (1 - \tau^{-1}) \left\| (\boldsymbol{X}^{(t+1/2)} - \boldsymbol{Y}^{(t)})(\mathbf{I} - \gamma_t\boldsymbol{W}) + \gamma_t(\boldsymbol{X}^{(t+1/2)} - \overline{\boldsymbol{X}}^{(t+1/2)})(\boldsymbol{W} - \mathbf{I}) \right\|_F^2$$

$$\leq (1 - \tau^{-1})(1 + \alpha_2)\|\mathbf{I} - \gamma_t\boldsymbol{W}\|_2^2 \left\| \boldsymbol{X}^{(t+1/2)} - \boldsymbol{Y}^{(t)} \right\|_F^2 + (1 - \tau^{-1})\gamma_t^2\left(1 + \alpha_2^{-1}\right)\|\mathbf{I} - \boldsymbol{W}\|_2^2 \left\| \boldsymbol{X}^{(t+1/2)} - \overline{\boldsymbol{X}}^{(t+1/2)} \right\|_F^2$$

$$\leq (1 - \tau^{-1})(1 + \alpha_2)(1 - \gamma_t\lambda_n)^2 \left\| \boldsymbol{X}^{(t+1/2)} - \boldsymbol{Y}^{(t)} \right\|_F^2 + (1 - \tau^{-1})\gamma_t^2\left(1 + \alpha_2^{-1}\right)(1 - \lambda_n)^2 \left\| \boldsymbol{X}^{(t+1/2)} - \overline{\boldsymbol{X}}^{(t+1/2)} \right\|_F^2.$$

$$\tag{67}$$

Combining (64) and (65),

$$\mathbb{E}\left[ (1 + \theta^{-1}) \left\| \boldsymbol{Z}^{(t+1)} - \overline{\boldsymbol{Z}}^{(t+1)} \right\|_F^2 + \left\| \boldsymbol{X}^{(t+1/2)} - \boldsymbol{Y}^{(t+1)} \right\|_F^2 \right]$$

$$\leq (1 + \theta^{-1})\max\{A(\gamma_t), B(\gamma_t)\} \left( \left\| \boldsymbol{X}^{(t+1/2)} - \overline{\boldsymbol{X}}^{(t+1/2)} \right\|_F^2 + \left\| \boldsymbol{X}^{(t+1/2)} - \boldsymbol{Y}^{(t)} \right\|_F^2 \right)$$

$$+ (1 + \theta)\eta_t^2 \left\| \nabla\boldsymbol{F}(\boldsymbol{X}^{(t+1)}, \xi^{(t+1)}) + \boldsymbol{\Phi}^{(t+1)} \right\|_F^2, \tag{68}$$

where

$$A(\gamma_t) = (1 + \alpha_1)\left(1 - \gamma_t\tilde{\lambda}\right)^2 + (1 - \tau^{-1})\gamma_t^2\left(1 + \alpha_2^{-1}\right)(1 - \lambda_n)^2,$$

$$B(\gamma_t) = \left(1 + \alpha_1^{-1}\right)\gamma_t^2\left(1 - \tilde{\lambda}\right)^2 + (1 - \tau^{-1})(1 + \alpha_2)(1 - \gamma_t\lambda_n)^2. \tag{69}$$

Comparing (68) with our goal in (32), it remains to show that $\max\{A(\gamma_t), B(\gamma_t)\} \leq 1 - \omega$. We bound the terms $A(\gamma_t), B(\gamma_t)$ as follows.

**Proof for** $A(\gamma_t) \leq 1 - \omega$: Let $\alpha_1 = \gamma_t \tilde{\lambda}/2$ and $\alpha_2 = 1/(2\tau)$. Following (Koloskova et al., 2019, Eqs. (20)–(24)),

for any $\kappa \in [0, 1]$, we have

$$A(\gamma_t) \leq 1 - \kappa \frac{\tilde{\lambda}^2}{8(1 - \lambda_n)^2 \tau + \tilde{\lambda}^2}. \tag{70}$$

Setting $\kappa = \frac{8(1-\lambda_n)^2\tau + \tilde{\lambda}^2}{8\tau}$ in (70), we have $A(\gamma_t) \leq 1 - \frac{\tilde{\lambda}^2}{8\tau} = 1 - \omega$, where the last equation follows from $\omega = \frac{\tilde{\lambda}^2}{8\tau}$.

**Proof for** $B(\gamma_t) \leq 1 - \omega$: Leveraging the inequality $(1 - x)(1 + x/2) \leq 1 - x/2, \forall x > 0$, we have

$$
\begin{aligned}
B(\gamma_t) &= \gamma_t^2(1 - \tilde{\lambda})^2 \left(1 + \frac{2}{\gamma_t \tilde{\lambda}}\right) + (1 - \tau^{-1})\left(1 + \frac{1}{2\tau}\right)(1 - \gamma_t \lambda_n)^2 \\
&\overset{\gamma_t \leq 1}{\leq} (1 - \tilde{\lambda})^2 \left(\gamma_t + \frac{2\gamma_t}{\tilde{\lambda}}\right) + \left(1 - \frac{\tau^{-1}}{2}\right)(1 - \gamma_t \lambda_n)^2 \\
&\overset{\gamma_t \leq 1}{\leq} (1 - \tilde{\lambda})^2 \left(1 + \frac{2}{\tilde{\lambda}}\right)\gamma_t + \left(1 - \frac{\tau^{-1}}{2}\right)(1 + \gamma_t \lambda_n^2 - 2\gamma_t \lambda_n) \\
&\overset{-1 \leq \lambda_n \leq 1}{\leq} (1 - \tilde{\lambda})^2 \left(1 + \frac{2}{\tilde{\lambda}}\right)\gamma_t + \left(1 - \frac{\tau^{-1}}{2}\right)(1 + 3\gamma_t) \\
&\overset{\tau > 0}{\leq} (1 - \tilde{\lambda})^2 \left(1 + \frac{2}{\tilde{\lambda}}\right)\gamma_t + \left(1 - \frac{\tau^{-1}}{2}\right)1 + 3\gamma_t \\
&= 1 - \frac{1}{2}\left(\tau^{-1} - \frac{2\tilde{\lambda}^3 + 4 - 12\tilde{\lambda}}{\tilde{\lambda}}\gamma_t\right). \tag{71}
\end{aligned}
$$

We consider the following two cases, depending on the value of $\tilde{\lambda} = 1 - |\lambda_2| \in [0, 1]$.

- If $\tilde{\lambda} \geq \frac{\sqrt{41}-5}{4} \approx 0.351$, one can verify that the cubic function $\tilde{\lambda}^2 + 2\tilde{\lambda}^3 + 4 - 12\tilde{\lambda} \leq 0$. Consequently, the last term in (71) is non-positive, yielding $B(\gamma_t) \leq 1 - \frac{1}{2\tau}$. Meanwhile, we have $\tilde{\lambda} \leq 1$ since $|\lambda_2| \leq 1$ and $\tau \geq 1$ from the definition in (6). This implies $B(\gamma_t) \leq 1 - \frac{1}{2\tau} \leq 1 - \frac{\tilde{\lambda}^2}{8\tau} = 1 - \omega$.

- If $\tilde{\lambda} < \frac{\sqrt{41}-5}{4} \approx 0.351$, we have $\tilde{\lambda}^2 + 2\tilde{\lambda}^3 + 4 - 12\tilde{\lambda} > 0$. When $\gamma_t \leq \frac{\tilde{\lambda}}{\tau(\tilde{\lambda}^2 + 2\tilde{\lambda}^3 + 4 - 12\tilde{\lambda})}$, it follows from (71) that

$$B(\gamma_t) \leq 1 - \frac{\tilde{\lambda}^2}{2\tau(\tilde{\lambda}^2 + 2\tilde{\lambda}^3 + 4 - 12\tilde{\lambda})}. \tag{72}$$

To prove $B(\gamma_t) \leq 1 - \omega$, it remains to verify if the following inequality holds:

$$\frac{\tilde{\lambda}^2}{2\tau(\tilde{\lambda}^2 + 2\tilde{\lambda}^3 + 4 - 12\tilde{\lambda})} \geq \frac{\tilde{\lambda}^2}{8\tau}. \tag{73}$$

Note that we have $\tilde{\lambda}^2 + 2\tilde{\lambda}^3 + 4 - 12\tilde{\lambda} \leq 4 - 3\tilde{\lambda} \leq 4$ since $0 \leq \tilde{\lambda} \leq 1$. Therefore, (73) holds.

### A.6.3 PROOF OF LEMMA 4

Let $m = \frac{4c(a+2\epsilon)}{ac + 4(c-2)\epsilon}$. We will prove Lemma 4 by induction. Suppose $r_t \leq m\frac{\eta_t^2 A}{c^2}$. For $t + 1$,

$$r_{t+1} \leq (1 - c/2)m\frac{\eta_t^2 A}{c^2} + \frac{2}{c}\eta_t^2 A \leq \frac{\eta_t^2 A}{c^2}\left((1 - c/2)m + 2c\right). \tag{74}$$

It is sufficient to prove

$$
\begin{aligned}
&\left((1-c/2)m+2c\right)\eta_t^2 \le m\eta_{t+1}^2\\
\Leftrightarrow\,&m(t+a)^{2\epsilon}-(t+a+1)^{2\epsilon}\left((1-c/2)m+2c\right)\ge 0\\
\Leftrightarrow\,&m-(1+\frac{1}{t+a})^{2\epsilon}\left((1-c/2)m+2c\right)\ge 0\\
\Leftrightarrow\,&\left(1+\frac{1}{t+a}\right)^{2\epsilon}\left((1-\frac{c}{2})m+2c\right)\le m
\end{aligned}
$$

Define a function $g(x)=(1+x)^{2\epsilon}-1-4\epsilon x$ for $x\in[0,1)$. Since $2\epsilon-1\le 1$,

$$
g'(x)=2\epsilon(1+x)^{2\epsilon-1}-4\epsilon\le 0. \tag{75}
$$

Therefore, $g(x)\le g(0)=0$, implying that $\left(1+\frac{1}{t+a}\right)^{2\epsilon}\le 1+\frac{4\epsilon}{t+a}$. Therefore, it is sufficient to prove

$$
\left(1+\frac{4\epsilon}{t+a}\right)\left(\left((1-\frac{c}{2}\right)m+2c\right)\le m \Leftarrow \left(1+\frac{4\epsilon}{a}\right)\left(\left((1-\frac{c}{2}\right)m+2c\right)\le m
$$

A simple calculation shows that it is sufficient to have $m\ge\frac{4c(a+2\epsilon)}{ac+4(c-2)\epsilon}$, which is true from the definition of $m$.

### A.6.4 PROOF OF PROPOSITION 2

The proof of Proposition 2 is similar to that in Appendix A.6.1. For any $\alpha>0$, we have

$$
\begin{aligned}
\mathbb{E}\left[\left\|\boldsymbol{X}^{(t+1)}-\boldsymbol{Y}^{(t)}\right\|_F^2\right] &= \mathbb{E}\left[\left\|\boldsymbol{Z}^{(t+1)}-\boldsymbol{Y}^{(t)}-\eta_t\left(\nabla\boldsymbol{F}(\boldsymbol{X}^{(t)};\xi_t)+\Phi^{(t+1)}\right)\right\|_F^2\right]\\
&\le (1+\alpha^{-1})\eta_t^2\mathbb{E}\left[\left\|\nabla\boldsymbol{F}(\boldsymbol{X}^{(t)};\xi_t)+\Phi^{(t+1)}\right\|_F^2\right]+(1+\alpha)\mathbb{E}\left[\left\|(1-\gamma_t)\boldsymbol{X}^{(t)}-\boldsymbol{Y}^{(t)}+\gamma_t\boldsymbol{Y}^{(t)}\boldsymbol{W}\right\|_F^2\right]\\
&\le 2(1+\alpha^{-1})(2G^2+2\sigma^2+\mu^2 d)n\eta_t^2+(1+\alpha)\mathbb{E}\left[\left\|(\boldsymbol{I}-\gamma_t\boldsymbol{W})(\boldsymbol{X}^{(t)}-\boldsymbol{Y}^{(t)})+\gamma_t\boldsymbol{X}^{(t)}(\boldsymbol{I}-\boldsymbol{W})\right\|_F^2\right].
\end{aligned}
\tag{76}
$$

We see that the expression in (76) is identical to (60). Following the same steps in Appendix A.6.1, we have Proposition 2.

## B EXPERIMENTAL RESULTS

### B.1 FEDERATED LEARNING OVER MNIST

We simulated 10 nodes within a fully connected network. This corresponds to the federated learning setup except we do not assume the existence of a central aggregator. The mixing matrix $\boldsymbol{W}=\boldsymbol{1}\boldsymbol{1}^T/n$ and $\gamma_t=1$. The DNN we are training consists of three fully connected layers with total $d=669,706$ parameters. The training data is distributed in a highly heterogeneous fashion by following Konečný et al. (2016), where each node has data corresponding to exactly two classes/labels with 60K/10K training/testing data in total. The consensus aggregation is performed in a synchronous manner, i.e., at iteration $t$ all the nodes simultaneously perform the aggregation step by (4). Additionally, to validate the efficacy of MALCOM-PSGD we use a constant learning rate ($\eta_t=.2$) and consensus step size ($\gamma_t=1$). For the error free baseline we use the standard decentralized SGD algorithm. To generate Figure 1

### B.1.1 RING-LIKE TOPOLOGY

**Simulation setup for the partially connected network**. We assume a network topology as illustrated in Figure 4 and we have also included its corresponding mixing matrix. This network topology is a neighborhood ring topology. That is, there are 4 neighborhoods connected together by nodes 0,2,4,6 and 8. Besides the change in network topology and mixing matrix we use the same data

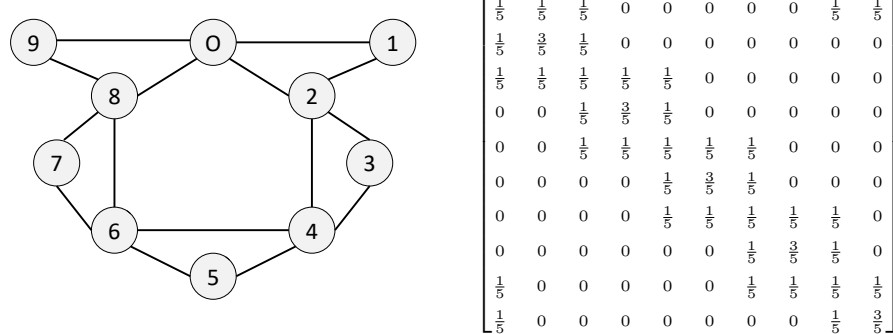

Figure 4: Left: The Ring-Like network topology. Circles denote the devices and edges denote connection links, where self-loops are omitted in the plot for brevity. Right: The corresponding mixing matrix $\mathbf{W}$.

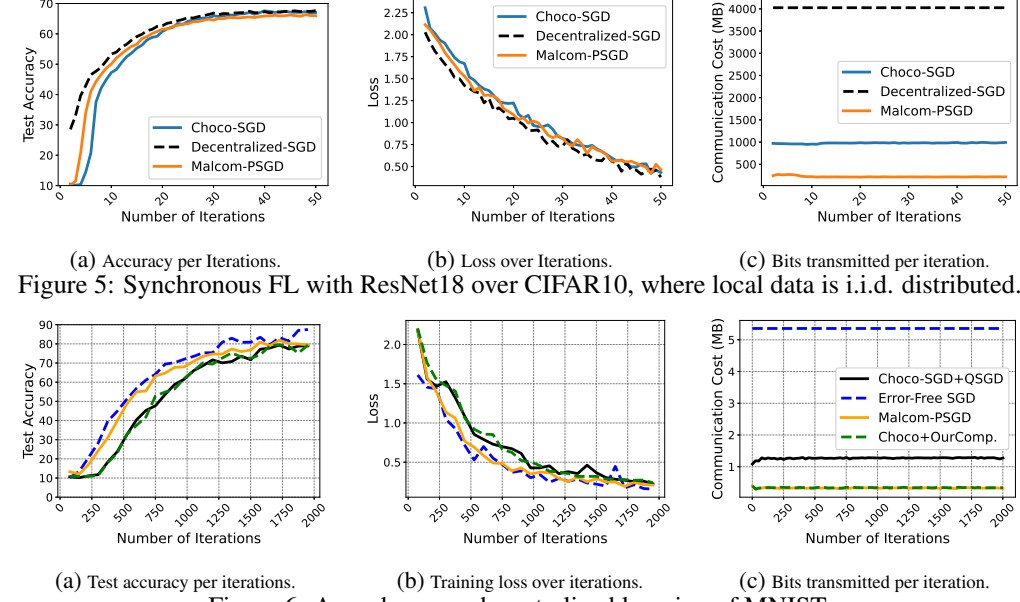

(a) Accuracy per Iterations.   (b) Loss over Iterations.   (c) Bits transmitted per iteration.

Figure 5: Synchronous FL with ResNet18 over CIFAR10, where local data is i.i.d. distributed.

(a) Test accuracy per iterations. (b) Training loss over iterations. (c) Bits transmitted per iteration.

Figure 6: Asynchronous decentralized learning of MNIST.

distribution, DNNs, and training/testing split as B.1. We utilize a constant learning rate ($\eta_t = .2$) and consensus step size ($\gamma_t = 1$).

For the sake of completeness we also have also plotted the number of total bits versus accuracy. See Figure 10

## B.2 EFFECTS OF SPARSIFICATION

The network and DNN setup is identical to B.1 accept we only consider MALCOM-PSGD and CHOCO-SGD+Our Compression Scheme. We vary the level of thresholding which corresponds to $\mu$ in (7).

## B.3 RESNET18 OVER CIFAR10

We simulate an FL network with 10 nodes where each node uses the ResNet18 model He et al. (2016) with an i.i.d. distribution of $50,000$ CIFAR10 training samples. Aggregation and communication protocols are the same as those in the above experiments with the MNIST dataset. The results are presented in Fig. 5. We observe similar results to that in Fig. 1, except that MALCOM-PSGD converges faster than CHOCO-SGD. Notably, MALCOM-PSGD reduces communication costs from the error-free baseline and CHOCO-SGD by $95\%$ and $78\%$, respectively.

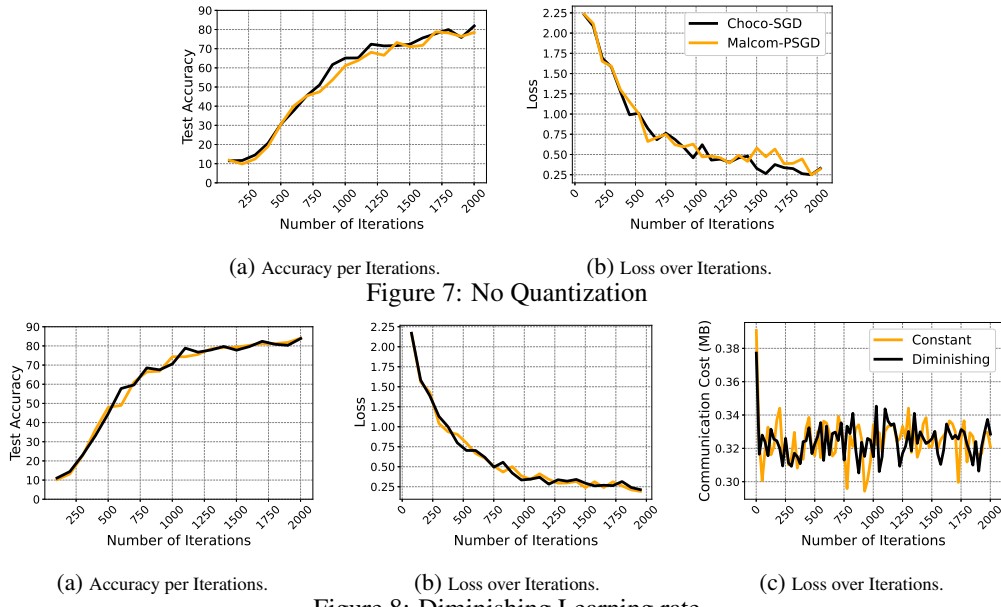

(a) Accuracy per Iterations.          (b) Loss over Iterations.

Figure 7: No Quantization

(a) Accuracy per Iterations.     (b) Loss over Iterations.     (c) Loss over Iterations.

Figure 8: Diminishing Learning rate

## B.4   ASYNCHRONOUS DECENTRALIZED LEARNING

This setup is identical to the FL setup above, except that we consider pairwise dynamic and asynchronous networking. At each iteration, we randomly pick two nodes for model sharing and aggregation over their communication link, resulting in a time-varying mixing matrix. The results are presented in Fig. 6. Similar to the synchronous case, both CHOCO-SGD and MALCOM-PSGD eventually recover the accuracy and loss of the error-free algorithm, but it should be noted that our algorithm converges at a slightly faster rate. This is because in the asynchronous setting error accumulation during aggregation is more significant due to the high data heterogeneity. Our findings indicate that MALCOM-PSGD reduces the communication cost by $94\%$ compared to the error-free baseline and by $74\%$ relative to CHOCO-SGD. This result highlights the efficiency of MALCOM-PSGD in the practical asynchronous setting, which represents numerous real-world situations, including Internet-of-Things (IoT) and device-to-device (D2D) networks.

## B.5   NO COMPRESSION

We assume the same set up as the Asynchronous Decentralized Learning experiment as described in B.4 but we apply no compression and encoding. The goal of this experiment is to show that adding sparsity has a marginal effect on the test accuracy and training loss as illustrated in Figure 7.

## B.6   DIMINISHING VS CONSTANT LEARNING RATE.

We assume the asynchronous set up as above but consider two different learning rates. For the constant learning rate we have $\eta = .2$ where as for the diminishing learning rate we follow (10): $\eta_t = \frac{1}{L(t+a)^\epsilon}$ setting $\epsilon = 1/2$, $L = .2$, and $\alpha = 200$. The results are summarized in Figure 8 and indicate that with properly chosen constants, a diminishing and constant learning rate can achieve similar performance.

## B.7   BIT CONSTRAINED EXPERIMENTS

We simulate decentralized image classification over the MNIST dataset with the aforementioned DNN. Local model residuals are communicated over rate-constrained channels with the channel capacity of 39 MB/iteration and 1 MB/iteration for the synchronous and asynchronous systems, respectively. Fig. 9 plots the test accuracy and training loss under the stringent communication rate constraints. While CHOCO-SGD diverges due to excessive quantization error, MALCOM-PSGD capitalizes on more efficient communication and thus allows high-precision quantization.

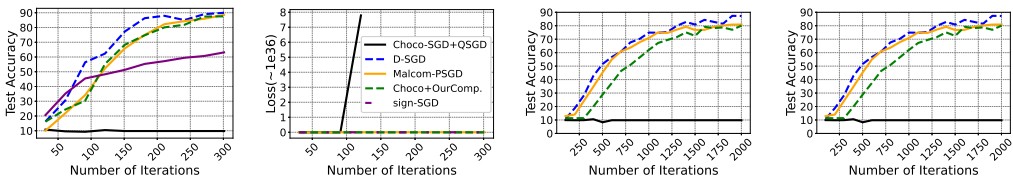

(a) Sync: Acc. versus Iter.   (b) Sync: Loss versus Iter.   (c) Async: Acc. versus Iter.   (d) Async: Loss versus Iter.

Figure 9: Decentralized learning under the communication rate constraints. The loss values of (b) and (d) are on the order of $10^{36}$ and $10^{24}$ respectively.

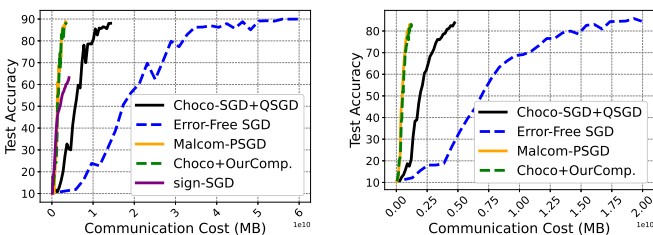

(a) Accuracy per Total Bits.   (b) Accuracy per Total Bits.

Figure 10: Bit Utilization vs Accuracy for Fully Connected and Ring Topology

