# OpenReview forum: "Malcom-PSGD: Inexact Proximal Stochastic Gradient Descent for Communication Efficient Decentralized Machine Learning"
_ICLR.cc/2024/Conference — Submitted to ICLR 2024_

### Official Review · Reviewer_ckRP · 2023-10-21

**Soundness:** 3 good
**Presentation:** 2 fair
**Contribution:** 2 fair
**Rating:** 5
**Confidence:** 4

**Summary:**

The paper focused on the communication bottleneck in decentralized ML problems. The authors provided a method named MALCOM-PSGD to reduce communication cost, which integrates the model sparsification and gradient compression. An $O(1/\sqrt{t})$ rate was established to guarantee the convergence of the proposed algorithm. Also the authors used numerical experiments to validate the performance.

**Strengths:**

The paper addresses the challenge of overcoming communication bottlenecks in the context of decentralized machine learning, with a particular focus on training large-scale, over-parameterized neural network models. The authors provided theoretical analysis and numerical experiments to validate the performance of the proposed algorithm.

**Weaknesses:**

The paper studied the traditional communication bottleneck issue in decentralized ML problems. The proposed algorithm appears to amalgamate two pre-existing techniques, exhibiting rather modest advancements. Moreover, the theoretical analysis appears to follow a standard course, and there is room for enhancing the experimental aspects.

**Questions:**

1. Can authors provide more explanation for the reason of combining model sparisificaion and gradient compression? Though both of them can improve communication efficiency, can the combination achieve `1+1>2` improvement? Also, is there any additional challenge from either implementation or theoretical analysis caused by the combination?

2. If we look at the convergence rate in Theorem 2, it is w.r.t. to $\mathcal{F}$ instead of $F$. To give a fair comparison, can the result be extended to  $F$?

3. After adding the model sparisificaion,  how will the model generalization be changed (gain or loss) in either theoretical way or numerical experiments? This part remains very unclear.

4. There are many other communication efficient methods, such as local updates/signSGD, why don't compare with them at least in the numerical experiments? I feel the numerical experiments can include many other communication efficient decentralized optimization methods.

5. From the experiments, the improvement w.r.t. test accuracy seems not very signification. Any analysis on it?

---

> ### Author Response · Authors · 2023-11-21
> **Response Pt 1**
>
> We thank the reviewer for the comments. We have organized our response as follows: We have  responded to your comments in the order they were given starting with the weaknesses comment.
>
> 1.
>     **Reply**: In this revision, we have provided a more detailed
>     exposition on the combined effects of model sparsification and
>     gradient compression. Additionally, we have delved deeper into the
>     challenges in our analysis and algorithm design. Furthermore, the
>     numerical results have been enriched with comparisons of our method
>     against an additional baseline algorithm. We have also incorporated
>     extra simulation results to illustrate the enhancement of the
>     communication bit rate from model sparsification. Please refer to
>     the replies below for more details.
>
> 2.
>     **Reply**: In our design, model sparsification is anticipated to
>     yield sparse model residuals as training progresses. This fact,
>     combined with the convergence of our algorithm that naturally
>     reduces the entropy of the quantization output, will reduce the
>     number of bits required for representing the input values, as long
>     as the compression scheme takes advantage of this. To specifically
>     highlight the improvement, we have added the following changes to
>     the manuscript:
>
>     -   We specialize the entropy rate with a concrete example, where
>         the model residuals have an i.i.d. Laplace distribution. Under
>         this assumption and a constant quantization precision, we show
>         that the communication bits decrease to a constant with the
>         training iteration $t$ on the order of $\mathcal{O}(-d\log t)$;
>         see Section 5 and Appendix A.3.
>
>     -   In Appendix A.3, we have added a table to compare the
>         communication bit rate of our method with QSGD \[Ref 5\] and
>         signSGD \[Ref 3\]. We show analytically that our method archives
>         compression that scales with the entropy of the quantization
>         output and the order of the model.
>
>     -   Section 6 includes a new plot, Fig. 3, which examines training
>         performance and total communication bits across 300 training
>         iterations for a fixed compression scheme and varying levels of
>         the $\ell_1$-regularization penalty $\mu$ defined in Eq. (2).
>         Here, a larger $\mu$ leads to sparser local models, while
>         $\mu=0$ indicates no sparsification. The orange curve in Fig.
>         3(c) shows that the optimal communication cost is achieved with
>         a non-zero $\mu=10^{-6}$. This setting also maintains comparable
>         training accuracy to the unsparsified model. The observation
>         underscores the importance of model sparsification in
>         decentralized learning: *when combined with gradient
>         compression, it further boosts communication efficiency without
>         compromising model accuracy.*
>
>     Although combining model sparsification and gradient compression
>     yields favorable improvements, it significantly complicates the
>     theoretical analysis on the convergence of the proposed algorithm.
>     The $\ell_1$-regularization for sparsification results in a
>     non-smooth objective, and the quantization errors that accumulate
>     over training iterations need precise quantification in the
>     convergence proof. Specifically, the non-smoothness precludes the
>     directly application of existing analyses, such as that in \[Ref
>     4\], which rely on leveraging the smooth properties to expand the
>     consensus objective value in each iteration. Moreover, the
>     intertwined quantization error, compounded by the non-convex nature
>     of the loss function, hinders straightforward computation of the
>     consensus gap in terms of the overall objective, which is commonly
>     achieved by using convexity properties in the existing work such as
>     \[Ref 6\]. Finally, the random quantization error in each training
>     iteration results in a generally non-monotonic objective value. This
>     contrasts with the scenario in error-free decentralized proximal
>     SGD, where the boundedness of the solution can be established, as
>     demonstrated in \[Ref 6, Lemma 17\]. To effectively address this
>     challenge, we have to incorporate an additional assumption as in
>     Assumption 4.
>
>     Our work provides a novel analysis for the convergence of
>     decentralized proximal SGD in the context of non-convex non-smooth
>     optimization. We believe that our findings advance the theoretical
>     understanding in this area, not to mention that empirically it shows
>     a $75\%$ gain in communication cost over the state-of-the-art.

---

> > ### Author Response · Authors · 2023-11-21
> > **Response Pt 2**
> >
> > 3.
> >     **Reply**: Please note that our decentralized optimization system is
> >     specifically designed to minimize the $\ell_1$-regularized loss
> >     objective $\mathcal{F}$ defined in Eq. (2), rather than the
> >     unregularized loss $F$. Consequently, it is pertinent to assess the
> >     performance of our algorithm in relation to $\mathcal{F}$.
> >
> >     In terms of numerical results, we note that all the baseline
> >     algorithms used in Section 6 are tailored for decentralized SGD of
> >     the unregularized (and thereby smooth) loss function $F$. As such,
> >     for a uniform comparison, we evaluate the performance of all the
> >     algorithms by *the unregularized loss* $F$. This metric, while not
> >     ideally suited to our method (since our solution is optimized with
> >     respect to $\mathcal{F}$ and thus is generally sub-optimal for $F$),
> >     still demonstrates that our method attains a comparable training
> >     loss to the baselines and at the same time significantly reduces the
> >     communication cost.
> >
> >     As a final note, we highlight that our analytical results can also
> >     be extended to systems that do not involve model sparsification.
> >     This is achievable by setting the $\ell_1$-regularization penalty
> >     $\mu$ to zero in our analysis, which corresponds to the minimization
> >     of $F$ using decentralized SGD without the proximal step. Under this
> >     setup, Theorem 2 indicates that the convergence order would remain
> >     consistent with that observed in the regularized case.
> >
> >     In the revised manuscript, we have Footnote 1 on Page 3 to clarify
> >     this point:
> >
> >     \"Since our optimization design is specifically aimed at minimizing
> >     the regularized loss, $\mathcal{F}$, our analysis will predominantly
> >     concentrate on the convergence with respect to $\mathcal{F}$, rather
> >     than the unregularized loss $F$.\"
> >
> >     We have also noted in Section 6 that our numerical results are
> >     evaluated using unregularized loss.
> >
> > 4.
> >     **Reply**: It has been widely recognized that model sparsification
> >     can enhance model generalization and avoid overfitting \[Ref 1\],
> >     but this also remains a timely research direction \[Ref 2\].
> >     However, we note that the focus of this work is on improving
> >     communication efficiency for the decentralized training of a
> >     predetermined learning model. While the exploration of model
> >     sparsification and gradient compression to model generalization is
> >     undoubtedly an intriguing topic, it lies outside the scope of this
> >     study and demands extensive dedicated research. We will explore this
> >     direction in future work and hope your understanding with this
> >     matter.
> > 5.
> >     **Reply**: In the revised manuscript, we have incorporated signSGD
> >     from \[Ref 3\] as a baseline for comparison in Fig. 1. The results
> >     in Fig. 1(c) shows that our method (orange curves) incurs a similar
> >     communication bit cost compared to signSGD (solid purple curves).
> >     However, our method excels in achieving more accurate model
> >     aggregation, leading to a substantial improvement in training loss
> >     and accuracy. In contrast, signSGD suffers from significant
> >     quantization errors due to its low quantization precision.
> >
> > 6.
> >     **Reply**: Please note that the primary goal of our study is to
> >     *enhance the communication efficiency* for decentralized learning.
> >     In our numerical results, we assess the performance of difference
> >     algorithms based on two criteria over a fixed number of training
> >     iterations: 1) training loss and test accuracy, and 2) communication
> >     efficiency, measured by the total communication bits required during
> >     training. For instance, examining the results presented in Fig. 1,
> >     our method (orange curves) is compared with the baseline CHOCO-SGD
> >     \[Ref 4\] combined with the compression scheme QSGD from \[Ref 5\]
> >     (dashed blue curves). It shows that our method achieves comparable
> >     testing accuracy with this baseline as they adopt the same
> >     quantization precision. However, our method reduces the
> >     communication bits by more than $75\%$ compared to the baseline, as
> >     demonstrated in Fig. 1(c). Another comparison, this time between our
> >     method (orange curves) and signSGD from \[Ref 3\] (solid purple
> >     curves) in the same figure, shows that while their communication
> >     costs are similar, our method attains more precise model
> >     aggregation, leading to a substantially faster convergence rate.
> >
> >     In Appendix B, we have shown another comparison in Fig. 9, where we
> >     fix for both algorithms the same communication bit budget ($1$ MB
> >     per iteration). It shows that the baseline algorithm, namely
> >     CHOCO-SGD, diverges, while we still maintain a good performance in
> >     terms of training accuracy.

---

> > > ### Author Response · Authors · 2023-11-21
> > > **Response Pt 3**
> > >
> > > References:
> > >
> > > [Ref 1]: Michael C Mozer and Paul Smolensky. Skeletonization: A technique for
> > > trimming the fat from a network via relevance assessment. In *Advances
> > > in Neural Information Processing Systems (NeurIPS)*, 1988.
> > >
> > > [Ref 2]: Brian R. Bartoldson, Ari S. Morcos, Adrian Barbu, and Gordon
> > > Erlebacher. The Generalization-Stability Tradeoff In Neural Network
> > > Pruning. In *Advances in Neural Information Processing Systems
> > > (NeurIPS)*, 2020.
> > >
> > > [Ref 3]: Jeremy Bernstein, Yu-Xiang Wang, Kamyar Azizzadenesheli, and Anima
> > > Anandkumar. SignSGD: Compressed optimisation for non-convex problems. In
> > > *Proceedings of the 35th International Conference on Machine Learning*,
> > > volume 80, pp. 560--569, July 2018.
> > >
> > > [Ref 4]: Anastasia Koloskova, Tao Lin, Sebastian U Stich, and Martin Jaggi.
> > > Decentralized deep learning with arbitrary communication compression. In
> > > *International Conference on Learning Representations*, volume 130, pp.
> > > 2350--2358, 2021.
> > >
> > > [Ref 5]: Dan Alistarh, Demjan Grubic, Jerry Li, Ryota Tomioka, and Milan
> > > Vojnovic. QSGD: Communication-efficient SGD via gradient quantization
> > > and encoding. In *Advances in Neural Information Processing Systems*,
> > > volume 30, pp. 1709--1720, 2017.
> > >
> > > [Ref 6]: Jinshan Zeng and Wotao Yin. On nonconvex decentralized gradient
> > > descent. arXiv preprint arXiv:1608.05766, 2018.

---

> > > > ### Comment · Reviewer_ckRP · 2023-11-21
> > > > **Thanks for authors' response!**
> > > >
> > > > I would like to thank authors for their detailed response.
> > > >
> > > > I appreciate the authors includes more experimental comparison in the revised paper. In Figure 1-3, the authors compared the methods w.r.t. iteration. But I think the main advantage of this work is communication efficiency. So it would be better to compare the methods w.r.t. communicated bits.
> > > >
> > > > For the theoretical part, I would like to hold my opinion that it would be better to provide a convergence on the unregularized loss so that it is comparable with other algorithms, which will definitely make the work more completed.
> > > >
> > > > I appreciate the authors effort and raise my score from 3 to 5.

---

> ### Author Response · Authors · 2023-11-21
> **Follow up Response**
>
> Thank you for reading our response and responding so quickly and for re-evaluating your score.  We feel your comments have improved the manuscript.  We have a short follow-up to your reply.
>
> 1. We have added this Figure for the Ring and Fully connected topology to the Appendix(Appendix B.1 Figure 10). Additionally, there are 4 plots in Appendix B.7 (Figure 9), which show the effects of fixing the number of bits per iteration. Under this setting, we demonstrate that our algorithm still achieves good accuracy while Choco-SGD diverges. We should also acknowledge that in Figures 1-2 we have already taken Choco-SGD to its limits. That additional bit reductions cause substantial performance degradation.
>
> 2. We recognize the reviewer’s interest in understanding how our approach theoretically behaves with respect to an unregularized loss, as considered in the baseline algorithms. However, it is important to note that our methodology diverges fundamentally in this aspect. We are minimizing a different problem than the state of the art, and thus our convergence analysis should be with respect to our objective function. Furthermore, it is a standard practice in the literature to provide convergence in regard to the regularized objective value. For example [Ref 6],[Ref 7], and [Ref 8] all prove convergence with regard to their composite objective.
>
> New References:
>
> [Ref 7]: Xiao, Lin and Zhang, Tong. A Proximal Stochastic Gradient Method with Progressive Variance Reduction. SIAM Journal on Optimization (2014)
>
> [Ref 8]: Zhize Li and Jian Li. A simple proximal stochastic gradient method for nonsmooth nonconvex optimization. In Proceedings of the 32nd International Conference on Neural Information Processing Systems (NIPS'18) (2018)

---

### Official Review · Reviewer_4xpx · 2023-10-31

**Soundness:** 3 good
**Presentation:** 3 good
**Contribution:** 3 good
**Rating:** 6
**Confidence:** 3

**Summary:**

This work proposes MALCOM-PSGD, a communication efficient decentralized optimization algorithm for smooth and non-convex objectives that combines several existing techniques for communication cost reduction.  The first one is to promote the sparsity of node model parameters using $\ell_1$ regularization and MALCOM-PSGD optimizes the resulting non-smooth objective using proximal SGD.  Furthermore, MALCOM-PSGD applies residual quantization and source coding techniques to reduce the communication cost between decentralized nodes at each communication round. This work gives detailed analysis of the communication cost and convergence rate of MALCOM-PSGD in the synchronous setting and, specifically, shows with properly chosen learning rate, MALCOM-PSGD is able to achieve a convergence rate of $O(\ln t / \sqrt{t})$ with $t$ iterations. Finally, experiments on optimizing DNNs in a federated learning setting demonstrate the fast convergence rate and low communication cost of MALCOM-PSGD compared to the SOTA decentralized baseline, in both synchronous and asynchronous settings.

**Strengths:**

- This work considers the important and interesting problem of communication efficient decentralized optimization. While communication efficient distributed and centralized optimization is well understood, this work explores the relatively less well understood decentralized settings.

- The paper presents MALCOM-PSGD in a well-organized and easy-to-follow way.

**Weaknesses:**

- In Section 1 the 3rd point under “our contributions”, it is stated “Our use of compressed model updates and differential encoding allows us to reasonably assume we are creating a structure within our updates that this encoding scheme is most advantageous under.” This sentence seems a bit confusing and it is unclear why this encoding scheme the most advantageous.

- The experiments section in the draft only considers the federated learning (FL) setting, which is essentially a distributed and centralized optimization, with a single choice of the mixing matrix W. Since the focus of the paper is in decentralized settings, it’d be better to present more results of MALCOM-PSGD under different mixing matrices W.

- Minor issue: in Eq.10, it should be $\sum_{i=1}^{n} \mathbb{E}[...] \leq \sum_{i=1}^{n} 2\mathbb{E}[...] + 2 ... $

- Minor issue: in Eq.16, it should be $\eta_k$ instead of $\eta_t$.

**Questions:**

- How does the mixing matrix W affect the convergence rate of MALCOM-SGD?

- Is $\omega$ in Theorem 2 the same as the one defined in Theorem 1?

- In Assumption 4, is $|| \mathbf{X}^{(t)}||$ the operator norm of $\mathbf{X}^{(t)}$ ? (unclear notation)

- Just as this work mentions in the introduction, sparsification and quantization are two major techniques for reducing communication cost in distributed optimization algorithms. If instead of quantizing the model residual (aka., line 3 in Algorithm 1) in MALCOM-PSGD, one sparsifies the model residual by applying, e.g., Rand-$k$ sparsification, can the current analysis of MALCOM-PSGD be extended to this new method?

- Is Eq.8 in Section 4 "bit rate analysis" used to compute the communication cost in the experiments?

- In Section 4 "bit rate analysis", it is stated "As the training converges, we expect a diminishing residual to be quantized in Step 3 of Alg. 1, resulting in a sparse quantized vector. " Does this imply as the training proceeds, the communication cost of the nodes at each round decreases? However, from the experiment results, e.g., plot (c) in Figure 1/2, the communication cost of MALCOM-PSGD remains the same across the communication rounds. Any comments on this?

---

> ### Author Response · Authors · 2023-11-21
> **Response Pt 1**
>
> We thank the reviewer for the comments. Each number corresponds to your comment in the order it was originally made.
>
> 1.
>     **Reply**: The superiority of our compression scheme results from
>     the exploitation of model sparsification and the shrinkage of model
>     residual values due to the convergence of our learning algorithm.
>     Specifically, our model sparsification set yields sparse model
>     residuals as training progresses. This fact, combined with the
>     convergence of our algorithm that naturally reduces the entropy of
>     the quantization output, reduces the number of bits required for
>     representing the input values in our compression scheme
>
>     We have updated the description in the Introduction for clarity.
>     Moreover, in this revised version, we have made the following
>     changes to demonstrate the improvement in communication efficiency
>     resulting from model sparsification and the low-entropy update.
>
>     -   We specialize the entropy rate with a concrete example, where
>         the model residuals have an i.i.d. Laplace distribution. Under
>         this assumption and a constant quantization precision, we
>         establish, based on our convergence result, that the
>         communication bits decrease to a constant with the training
>         iteration $t$ on the order of $\mathcal{O}(-d\log t)$; see
>         Section 5 and Appendix A.3.
>
>     -   In Appendix A.3, we have added a table to compare the
>         communication bit rate of our method with QSGD \[Ref 5\] and
>         signSGD \[Ref 3\]. We show analytically that our method archives
>         the compression scales with the entropy of the quantization
>         output and the order of the model.
>
>     -   Section 6 includes a new plot, Fig. 3, which examines training
>         performance and total communication bits across 300 training
>         iterations for a fixed compression scheme and varying levels of
>         the $\ell_1$-regularization penalty $\mu$ defined in Eq. (2).
>         Here, a larger $\mu$ leads to sparser local models, while
>         $\mu=0$ indicates no sparsification. The orange curve in Fig.
>         3(c) shows that the optimal communication cost is achieved with
>         a non-zero $\mu=10^{-6}$. This setting also maintains comparable
>         training accuracy to the unsparsified model. The observation
>         underscores the importance of model sparsification in
>         decentralized learning: *when combined with gradient
>         compression, it further boosts communication efficiency without
>         compromising model accuracy.*
>
> 2.
>     **Reply**: We have added additional numerical results with a
>     different choice of $\bf W$ in Fig. 2 of the manuscript. This
>     experiment considers a different graph topology shown in Fig. 4 from
>     Appendix B. Moreover, in Fig. 6 of Appendix B, we present another
>     simulation with a time-varying mixing matrix ${\bf W}_t$ over the
>     training iteration $t$, where each ${\bf W}_t$ represents a
>     sub-sampled partially connected network. In these experiments, we
>     have observed consistent results with the FL case: Our method
>     outperforms the state-of-the-art baselines in terms of communication
>     efficiency.
> 3.  **Reply**: We have corrected the minor issues in Eqs. (10) and (16).
>
> 4.
>     **Reply**: The second largest absolute eigenvalue of $\bf W$, i.e.,
>     $|\lambda_2|$, determines the value of the key parameter
>     $\omega=\frac{(1-|\lambda_2|)^2}{8\tau}$ in Theorem 1. In scenarios
>     where the graph is more connected, $|\lambda_2|$ tends to be small,
>     leading to a larger $\omega$ and thus a reduced consensus error
>     bound in Theorem 1. Furthermore, a larger $\omega$ also contributes
>     to a smaller optimality gap in Theorem 2, thereby leading to faster
>     convergence.
>
>     In the revised manuscript, we have explained this point under
>     Theorem 2, as
>
>     "For a decentralized learning network exhibiting higher
>     connectivity, the eigenvalue $|\lambda_2|$ of the mixing matrix
>     $\bf W$ is generally smaller. This results in a larger $\omega$ as
>     per Theorem 1, consequently leading to improved consensus and
>     convergence rates according to Theorems 1 and 2.\"
> 5.
>     **Reply**: Yes. We have clarified this in Theorem 2.
> 6.
>     **Reply**: The norm $||{\bf X}^{(t)}||$ refers to the Frobenius norm
>     $||{\bf X}^{(t)}||_F$. We have clarified this in Assumption 4.
>
> 7.
>     **Reply**: Yes. As shown in \[Ref 1, Lemma A.1\], Rand-$k$
>     sparsification satisfies Assumption 2 on Page 4, with the parameter
>     $\tau=d/k$, where $d$ is the size of the model. This implies that
>     our analysis is directly applicable to Rand-$k$. We have mentioned
>     this point under Assumption 2 in the revised manuscript.
>
>     [Ref 1]: Sebastian U Stich. Local SGD converges fast and communicates
>     little. In *International Conference on Learning Representations*,
>     2019.

---

> > ### Author Response · Authors · 2023-11-21
> > **Response Pt 2**
> >
> > 8.
> >     **Reply**: No. We note that the bit expression presented in the bit
> >     analysis section serves as an *upper bound* for the communication
> >     cost of our scheme. To provide a more accurate illustration of the
> >     communication cost, we empirically computed the *exact* number of
> >     bits in our simulations. We have clarified this in Section 6.
> >
> > 9.
> >     **Reply**: Yes. In Section 5, we have expanded our bit analysis to
> >     more effectively demonstrate the evolution of communication bits.
> >     Specifically, we establish in Eq. (18) that under a defined
> >     statistical model for model residuals and with a fixed quantization
> >     precision, the communication bits converge to a non-zero constant
> >     with high probability, at a decreasing rate of
> >     $\mathcal{O}(-d\log t)$. Here, $d$ is the size of the model, and $t$
> >     is the training iteration.
> >
> >     Regarding the numerical results, we apologize for the difficulty in
> >     discerning the trend of communication cost in the earlier figures
> >     due to poorly set y-axis margins. We have revised the figures in
> >     Section 6 and the Appendix to enhance their readability.
> >     Specifically, Fig. 8(c) in Appendix B.6 now clearly depicts the
> >     evolution of communication bits over the training iterations for our
> >     scheme. We observe that the required number of bits (the black
> >     curve) decreases as training progresses, eventually stabilizing
> >     around a constant level. The fluctuation in the plot results from
> >     the randomness arising from variations in quantization error and
> >     mini-batch data sampling. The trends observed in Fig. 8 align well
> >     with our analysis in Section 5.

---

### Official Review · Reviewer_zv9X · 2023-11-02

**Soundness:** 3 good
**Presentation:** 2 fair
**Contribution:** 2 fair
**Rating:** 5
**Confidence:** 4

**Summary:**

This paper aims to improve the communication efficiency of decentralized nonconvex optimization. In addition to compression that is heavily used recently, the authors also suggest to add an $\ell_1$ regularization to encourage model sparsity to help communication efficiency. The authors show that the convergence rate for consensus and objective convergence is matching the state-of-the-art. Moreover, the authors show in experiments that new method provides approximately 75% improvement in terms of communication costs.

**Strengths:**

The motivation of the paper, improving the communication efficiency for decentralized optimization and ML is definitely important. Moreover, the paper combines techniques from different areas including not only stochastic and decentralized optimization but also coding theory such as Elias coding and Golomb coding which is a positive. The paper does a decent job of comparing with the existing works (even though there are some unclear points about this that I touch on later). The experimental validation is helpful to show that the provided algorithm is a good candidate for achieving the main goal of the paper. The paper is also well-written and mostly polished.

**Weaknesses:**

Since the paper brings together many ideas, like stated in the beginning of Sect 3: "SGD, residual compression, communication and encoding, consensus aggregation and proximal optimization", it's at times difficult to distinguish in which aspects the work is different from the existing methods since sometimes both assumptions and problems are different than previous works. It is also not clear how significant the contribution is in addition to the existing techniques (which is not necessarily a reason for rejection since combination of existing tools can be interesting if the result is interesting enough, but this should be clarified much more).

For example, there is a decent comparison to Zeng & Yin (2018b) where the difference is handling the stochastic case. However, the comparison with more recent works, for example in Koloskova et al. 2021 is less clear. Is the main difference handling the $\ell_1$ regularizer? The authors mention a couple of times, for example right after Theorem 1 that "there are two key differences with Koloskova et al. 2021 for consensus bound: (1) decreasing step size instead of constant one (2) tighter upper bound in terms of C/\omega^3", with no further explanation. It is not clear why a decreasing step size is better or what the precise "tighter upper bound" is compared to previous work: what is the exact improvement in the bound? After I go look at Koloskova et al. 2021, I see that the "constant step size" in the existing paper depends on the final iterate $T$ whereas the decreasing step size in this paper depends on $t$, which I assume is the difference. Apart from not having to set $T$ in advance, what is the advantage of this? For the second part, tighter upper bound, it was less clear even after I had a quick look at Koloskova et al. (2021) (quick look since I ideally would prefer not to review other papers to be able to review one paper especially in such tight timelines as usual for ML conferences), the comparison is unclear because the bounds have different dependences. What corresponds to $C$ and $\omega$ in Koloskova et al. 2021 bound so that I can see what the improvement is?

Even more confusing is the difference on the assumptions. Rate of Koloskova et al. 2021 is on the gradient norm, which is standard for nonconvex optimization. I see that since the problem in this paper is regularized, gradient norm itself is not enough, but one can instead look at the prox-mapping norm. However, also surprisingly the rate in this paper in Theorem 2 is on the objective value, which is not standard for a nonconvex optimization problem. This points out to the difference on the assumptions, this current paper assumes coercivity, whereas Koloskova et al. 2021 does not (from my quick look again, please correct me if I am wrong). Unfortunately, there is no comment about this in the paper after Theorem 2, how come we can now get a guarantee on the objective value (a very strong guarantee for nonconvex optimization requiring very strong additional assumptions) whereas Koloskova et al. 2021 only gets gradient norm. What would we get by only assuming the standard assumptions such as Koloskova et al. 2021? Can we get a similar rate for the prox mapping?

On the same topic, I also had to have a quick look at Zeng & Yin (2018b) trying to understand the difference on assumptions. The eqs. (88), (89) that this paper uses in the middle of page 17, is from Lemma 21 in Zeng & Yin (2018b) that additionally assumes that the objective function is convex. This would explain the resulting rate in the objective value, whereas the current submission does not mention this. Can the authors explain if the estimations they use from Zeng & Yin (2018b) use convexity of the objective or not? As a result, does the current submissions use convexity (or any other additional assumptions) in some way or not?

Moreover, the paper mentioned in page 2 "our contributions" paragraph: "Our findings suggest that the gradient quantization process introduces notable aggregation errors ..... Making the conventional analytical tools on exact PGD inapplicable". Can you please clarify more? In my reading of the paper and the proofs, the analysis looks like a combination of Zeng & Yin (2018b) with Koloskova et al. (2021) and some tools from Alistarh et al. 2017 with Woldemariam et al. 2023 for encoding/decoding. If the authors claim that there are difficulties in combining these techniques, they should clearly state that. Even if there are not difficulties in combining techniques, this can also be fine if the result is strong enough. But this should be clarified by the authors.

The paper argues there is 75% communication improvement in practice, what about theory? Do the bounds predict any improvement? Moreover, what is the main sources of improvement in practice compared to choco-SGD? Is it using $\ell_1$ regularization leading to sparse solutions? If so, how to quantify this? Page 5 in paragraph "Source coding" mentions "intuitively" regularized problem produces sparse solutions, but can the authors provide a precise theoretical evidence for this? Moreover, the authors mention that the consensus aggregation  is different from Koloskova et al. 2021, since they use a scheme from Dimakis et al. 2010, is this also helping to improve the communication efficiency? These really need to be clarified.

Numerical results are a bit confusing. The paper solves the regularized problem whereas choco-SGD solves the unconstrained problem. How do the authors compare these two different methods solving different problems? Moreover, the authors say they use constant step size which is a bit disconnected from theory.

**Questions:**

- eq. (55) please provide a pointer to the definition of $\Phi^{(t+1)}$ as the sub gradient, for example after eq. (30). Also, after eq. (30) it calls $\Phi^{(t+1)}$ Subdifferential whereas it should be subgradient.

- Can you describe clearly Elias coding and Golomb coding used in Algorithm 2 (unfortunately many readers might be not familiar with coding theory) even if they are not essential for the purpose of the paper? Where do they come in to play in the analysis? Is it only the eq. (8) that is derived in the paper of Woldemariam et al. 2023? Also, for eq. (8) in Sect. 4 please provide a precise pointer in Woldemariam et al. 2023 where this result is proven so that a reader would know where to look. Also, please show clearly how Algorithm 2 fits within the main algorithm. In the "encoding" step of Algorithm 1, you may mention you call Algorithm 2 explicitly.

- Assumptions 3i is written in a bit confusing way, please consider writing it like $x_i \mapsto F_i(x_i)$ has $L_i$  as the Lipschitz constant for gradient so then it will be clear the sum is Lipschitz gradient  with the max of $L_i$. Please also provide more commentary about the coercivity in Assumption 3ii since it is not standard for nonconvex optimization and also different from existing works for example Koloskova et al. 2021.

- eq. (4) please explain better the difference of this scheme with Koloskova et al. 2021. Especially since the notations in the two papers are different, it is not easy for the reader to compare.

- footnote in page 4: What about theory? Does the theoretical results go through with asynchronous and time-varying network? If so, more justifications are needed.

- Second page says that Nesterov proposed proximal gradient descent whereas Nesterov in this paper points to earlier work (including a paper from 1981) for this "unaccelerated" PGD. Can you please correct the reference for proximal gradient descent?

- page.3 states "all prior analysis on convergence of quantized models rely on smoothness" as if this paper does not. But this paper also does, since all the proximal gradient methods do. They still use smoothness in a very central way and handle structural nonsmoothness with proximal operator. Better to be not misleading on this.

- It is not clear to me what is "inexact proximal gradient" referring to here. For example in the paper of Schmidt et al. 2011, inexactness is both on the gradient computation and proximal operator. Here the proximal operator seems to be exact, am I missing something? Is the inexactness due to compression and other techniques used for improving communication efficiency?

- Theorem 1: please point out to the definition of $\tau$ from Assumption 2 in Theorem 1 for improving readability.

- Right before eq. (58) the authors refer to some calculations in Koloskova et al. 2019 (eqs. (20)-(24)). Can you explicitly write these steps in the paper so that the reader will not have to go to another paper, get familiarized with their notation to come back, recall the notation of the current paper and then understand the steps?

- What can we get with the same set of assumptions as Koloskova et al. 2021 by not introducing more assumptions? This would probably be a rate on the prox-mapping.

---

> ### Author Response · Authors · 2023-11-21
> **Response Pt 1**
>
> We want to thank the review for their detailed comments on the
> manuscript. We understand the time commitment and difficulty in
> reviewing our paper within the time constraints. We have found your
> comments to be constructive and insightful and we have done our best to
> address them. We consider each of your paragraphs a single question for the formatting of the response.
>
> 1.
>     **Reply**: Thank you for highlighting the ambiguity. We have added a
>     few statements throughout the paper to, hopefully, clarify where we
>     are different and where we have added a contribution. Analytically
>     there was difficulty in combing the non-smoothness and compression
>     errors for convergence. We have edited the contribution section
>     stating that our contributions are as follows:
>
>     1.  Our most novel contribution is combining sparsifcation,
>         communicating the compressed residual and adopting the novel
>         compression scheme proposed in Woldemariam (2023), that is best
>         suited to leverage the low entropy of the resulting updates. The
>         whole is greater than the sum of its parts. On the other hand,
>         from the and analytical standpoint the heavy lifting comes from
>         the non-smooth and non-convex nature of our algorithm which
>         required a non-trivial convergence analysis. Numerically, we
>         demonstrate a 75% reduction in communication costs when compare
>         to the state of the art.
>
>     2.  Analytically We prove that MALCOM-PSGD converges in the
>         objective value for non-convex and smooth loss functions with a
>         given compression error. MALCOM-PSGD exhibits a convergence rate
>         of $\mathcal{O}(\ln{t}/\sqrt{t})$ and a consensus rate of
>         $\mathcal{O}(1/t)$ with diminishing learning rates and consensus
>         stepsizes, where $t$ represents the number of training rounds.
>         This analysis was complicated by the fact that our aggregation
>         scheme does not preserve the average of the iterates meaning we
>         could not directly apply Koloskova et al. (2021) and Zeng & Yin
>         (2018b).
>
>     Additionally, when we introduce our aggregation scheme (equation 4)
>     we have added the following sentence to clarify this is one of our
>     contributions: " This aggregation protocol is non-standard and is
>     thus one of our contributions.\" Additionally we add the following
>     line to Consensus Aggregation on page 5 "Even though it does not
>     preserve the average of the iterates and complicates the analysis,
>     this aggregation scheme was chosen because it reduces the error
>     accumulation caused by $Q(\bf{x})$\"
>
>     In regards to the compression operator we adopt the same assumptions
>     that Koloskova et al. (2021) uses. Meaning Assumption 2 is now:\
>     For any input ${\bf x}$,$Q({\bf x}):\mathbb{R}^d\rightarrow\mathbb{R}^{d}$
>     satisfies that $$\mathbb{E}[\|{Q({\bf x})-\bf x\|}^2]\leq (1-\frac{1}{\tau})\|{\bf x}\|^2, \qquad Q({\bf 0})={\bf 0},$$ where $\tau\in (0,1)$ is a constant representing the
>     expected compression error.\
>     Finally, in section 4 where we provide theorem 1 and 2, we have
>     added the following remark to clarify the differences in assumptions
>     between Koloskova et al. (2021), Zeng & Yin(2018b), and our own. We
>     have added: "Assumption 3(ii) is consistent with Zeng & Yin (2018b)
>     and under this condition, Assumption 4 requires that the loss values
>     produced in training iterations of our algorithm are finite. This
>     assumption can be reliably met with practical machine learning
>     solvers. We note that Assumption 4 is not required in Koloskova et
>     al. (2021) and Zeng & Yin (2018b), but is critical for our analysis
>     since we need it to bound the optimality gap. Koloskova et
>     al. (2021) is able to avoid this assumption by assuming a smooth
>     objective function while Zeng & Yin (2018b) avoids it by assuming
>     loss-less communication\".
> 2.
>     **Reply**:
>     Thank you for pointing out the inconsistency here, we agree that
>     this is an oranges to apples comparison so we have decided to remove
>     it from the manuscript. Since we have a non-smooth objective the
>     diminishing learning rate is required and thus makes a direct
>     comparison challenging. Importantly we have the same asymptotic
>     bound as Koloskova et al. (2021).

---

> > ### Author Response · Authors · 2023-11-21
> > **Response Pt 2**
> >
> > 3.
> >     **Reply:** Unfortunately, we did not succeed in utilizing the
> >     proximal mapping, because we were unable to bound the proximal
> >     mapping norm as it was unclear how to bound the effect of the lossy
> >     compression. Especially considering that we designed our aggregation
> >     technique to limit the effect of compression error boosting the
> >     impact of sparsity and thus bit efficiency. While this change was
> >     beneficial it complicated the analysis since we no longer preserve
> >     the average of the iterates meaning the standard analysis tools are
> >     no applicable.
> >
> >     In reagrds to your second point, we assume coercivity and Koloskova
> >     et al. (2021) does not. However, as already stated coercivity is
> >     required for our analysis since we need it to bound the optimally
> >     gap. That is $\|X^*-X^{(t)}\|\leq A$. Additionally we would like to
> >     note that lemma 17 in Zeng & Yin (2018b) states that the sequence of
> >     $\bf{x}^{(k)}$ is bounded. We cannot prove this lemma because we
> >     have the compression errors and hence why we introduce Assumption 4.
> > 4.
> >     **Reply**: Yes. The proof in Zeng & Yin (2018b) assumes convexity,
> >     while our proof does not. There is a critical step where our
> >     approach must diverge without the assumption of convexity.
> >     Specifically, Eq. (74) from the extended version of Zeng & Yin
> >     (2018b), available on https://arxiv.org/abs/1608.05766, uses
> >     convexity properties to bound the objective value. In contrast, our
> >     approach employs the smoothness property for this purpose,
> >     introducing an additional error term represented by
> >     $||{\bf X}^{(t)}-\overline {\bf X}^{(t)}||_F^2$ in Eq. (51) of our
> >     manuscript. We tackle this challenge by bounding the additional
> >     error terms using the result in Theorem 1 of the manuscript.
> > 5.
> >     **Reply:** There are 3 fundamental difficulties that we faced in the
> >     analysis that prevented us from directly applying Zeng & Yin (2018b)
> >     with Koloskova et al. (2021). We have the problems caused by the
> >     $\ell_1$ norm, error due to compression and the fact our aggregation
> >     scheme does not preserve the average of the iterates. As stated
> >     above and in the manuscript we cannot directly apply Zeng &
> >     Yin (2018) because they do not have any method for handling the
> >     errors caused by compression. One might assume then that we could
> >     combine Koloskova et al. (2021) with Zeng & Yin (2018), but this
> >     does not work since Koloskova et al. (2021)'s analysis requires that
> >     the aggregation scheme preserve the average of the iterates. Our
> >     aggregation scheme does not preserve the average of the iterates.
> >     Therefore, our convergence proof has 2 key differences from the Zeng
> >     & Yin (2018), we introduce Assumption 4, which we believe is a
> >     reasonable and realistic assumption for machine learning, and two we
> >     bound the distance from the average
> >     $\|{\bf X}^{(t)}-\overline{\bf{X}}\|_F$. Of course we utilized the
> >     relevant ideas and techniques in the literature, as does every
> >     researcher.

---

> > > ### Author Response · Authors · 2023-11-21
> > > **Response Pt 3**
> > >
> > > 6.
> > >     **Reply**: As we have hopefully already clarified, it is the
> > >     combination of the aggregation technique, $\ell_1$ induced
> > >     sparsifciation, and the compression scheme in Woldemariam et
> > >     al. (2023) that lead to a substantial improvement in the
> > >     communication efficiency. We have added Figure 3 to the manuscript
> > >     which provides empirical evidence of this fact. Figure 3 shows that
> > >     when Malcom-PSGD utilizes zero sparsification, we still utilize
> > >     fewer bits than Choco-SGD with our compression scheme. This implies
> > >     that our aggregation scheme independently aids communication
> > >     efficiency. The intuition is that since we utilize local values
> > >     instead of lossy reconstructions, we have smaller perturbations
> > >     caused by aggregation and thus tend to send fewer bits per
> > >     iteration. Furthermore Figure 3 indicates that there exist multiple
> > >     threshold values $\mu$ which reduces the communication costs
> > >     further.
> > >
> > >     Analytically it is not clear how one would show that our aggregation
> > >     technique aids the communication compression and encoding, we think
> > >     this is not possible. However we have provided analytical results
> > >     that provide an improvement on the bound based on the encoding
> > >     scheme and an analysis of how sparsity directly improves the
> > >     encoding. Please see Appendix A.3. Here we derive a bound for the
> > >     number of bits where we assume a Laplacian prior distribution
> > >     parameterized by $\rho_t$ where $\rho_t$ is diminishing, i.e. the
> > >     model is becoming more sparse. The analysis indicates that this
> > >     encoding scheme coupled with the model spasrification decreases the
> > >     number of bits on the order of $\mathcal{O}(-d\log(\Gamma))$, where
> > >     $d$ is the dimensionality of the model and $\Gamma$ is the number of
> > >     levels for quantization. Additionally, We have identified the
> > >     communication complexity and have provided a table in Appendix A.3.
> > >
> > >     The main source of improvement, which should also be clear from the
> > >     Figures\[1-3\], comes from the quantization and encoding mechanism.
> > >     We did additional simulations where we also gave Choco-SGD our
> > >     compression/endcoding scheme. Majority of the bit reduction comes
> > >     from the compression/encoding, but there are additional improvements
> > >     from model sparsfication and aggregation protocol.
> > >
> > >     In regards to the intuition on page 5, there was a mistake, as
> > >     stated in the paper the $\ell_1$ norm approximates the $\ell_0$ and
> > >     therefore solving a minimization problem where the $\ell_1$ norm is
> > >     included yields sparser solutions. We refer to Donoho (2006) as a
> > >     reference. Whether or not the $\ell_1$ norm is the best approach for
> > >     promoting sparsity in the context of decentralized learning, but
> > >     that is beyond the scope of this paper. We have edited this
> > >     statement to clearly state that the $\ell_1$ norm induces sparsity.
> > >
> > >     We also would like to clarify that we are not using any aggregation
> > >     scheme from Dimakis et al. 2010. We wanted to reference that
> > >     gossiping algorithms, which include our communication scheme, do
> > >     converge to some average, but in the original draft it suggested our
> > >     aggregation scheme directly came from Dimakis et al. 2010. We have
> > >     modified the text around equation 4 to clarify this.

---

> > > > ### Author Response · Authors · 2023-11-21
> > > > **Response Pt 4**
> > > >
> > > > 7.
> > > >
> > > >     **Reply**: We have changed the numerical results section and have
> > > >     provided new figures that better illustrate that Malcom-PSGD
> > > >     holistically improves the communication efficiency illustrating that
> > > >     the whole is greater than the sum of its parts.
> > > >
> > > >     Addressing your concern regarding the comparison of different models
> > > >     for solving different problems, we believe that evaluating both test
> > > >     accuracy and communication costs is a fair comparison. Each
> > > >     algorithm was given the same training data set with the same
> > > >     non-i.i.d distribution over the same networking topology. Both
> > > >     algorithms are trying to maximize testing accuracy via minimization
> > > >     of a loss function. Even though they use different loss functions we
> > > >     don't consider this to be problematic. Similarly, the motivation for
> > > >     using a constant learning rate was to demonstrate that Malcom-PSGD
> > > >     is a robust algorithm and will still converge in practice even with
> > > >     a very simple learning rate scheduler. Furthermore adopting a
> > > >     constant learning rate aids in the comparison between Malcom-PSGD
> > > >     and the baseline algorithms since we implemented all of them with
> > > >     constant learning rates. Finally, we did not want to give
> > > >     Malcom-PSGD an unfair advantage by giving it a more "tune-able\"
> > > >     learning rate. That is, we did not want our improvement on the state
> > > >     of the art to be the result of better hyper-parameter choice. But,
> > > >     for completeness and per your suggestion, we have added a simulation
> > > >     to Appendix B.6 that compares a diminishing learning rate to a
> > > >     constant learning rate. The results are presented in Figure 8 and
> > > >     they indicate that Malcom-PSGD achieves equivalent performance with
> > > >     a diminishing learning rate as with a constant one.
> > > > 8.  *eq. (55) please provide a pointer to the definition of as the sub
> > > >     gradient, for example after eq. (30). Also, after eq. (30) it calls
> > > >     Subdifferential whereas it should be subgradient.*
> > > >
> > > >     **Reply:** Good catch! This has been corrected.
> > > > 9.
> > > >     **Reply:** We have added statements describing Elias and Golomb
> > > >     coding to the Appendix. Although not used in the analysis in this
> > > >     paper, Elias and Golomb coding are used in part of the encoding
> > > >     scheme referenced. It is only equation (8), now equation (17), that
> > > >     is derived from Woldemariam et al. 2023, and we have now included a
> > > >     pointer to the equation number used in the paper. Equation (18)
> > > >     reflects additional analysis under the assumption that the weights
> > > >     are derived from a Laplacian pdf. We have also added a note in
> > > >     Algorithm 1 referring to the encoding scheme.
> > > >
> > > > 10.
> > > >     **Reply**: We have updated Assumption 3.ii to the notation you
> > > >     suggested. Additionally we have added a comment after Assumption 4
> > > >     which states that coercivity is utilized in Zeng & Yin (2018a) and
> > > >     we add a further explanation of why this assumption is needed for
> > > >     us. We include it here aswell:
> > > >
> > > >     "Assumption 3(ii) is consistent with Zeng & Yin (2018a) and under
> > > >     this condition, Assumption 4 additionally requires that the loss
> > > >     values produced in training iterations of our algorithm are finite.
> > > >     This assumption can be reliably met with practical machine learning
> > > >     solvers. We note that Assumption 4 is not required in Koloskova et
> > > >     al. 2021 and Zeng & Yin (2018a), but is critical for our analysis
> > > >     since we need it to bound the optimality gap. Koloskova et al. 2021
> > > >     is able to avoid this assumption by assuming a smooth objective
> > > >     function while Zeng & Yin (2018a) avoids it by assuming loss-less
> > > >     communication."
> > > > 11.
> > > >     **Reply**: Please see eq (4) we have updated it and included Choco-SGDs aggregation scheme in our notation as well.
> > > > 12.
> > > >     **Reply**: We considered an analysis of the asynchronous case to be
> > > >     beyond the scope of this paper. However we did provide an
> > > >     asynchronous implementation and experimentation because we wanted to
> > > >     demonstrate that Malcom-PSGD is robust and is applicable to both
> > > >     synchronous and asynchronous settings. However, we did move our
> > > >     asynchronous results to the appendix due to space constraints.
> > > > 13.
> > > >     **Reply**: Again, good catch! We really appreciate the detailed
> > > >     review you have provided. We have fixed this.
> > > > 14.
> > > >     **Reply**: Yes we agree that this is misleading. We believe we have
> > > >     clarified where non-smoothness is explicitly a problem for our
> > > >     analysis. We have removed this statement.
> > > > 15.
> > > >
> > > >     **Reply**: We agree this is confusing. By inexact we were referring
> > > >     to the error caused by compression. If we are accepted we will
> > > >     happily change the title to "MALCOM-PSGD: Compressed Proximal
> > > >     Stochastic Gradient Descent for Communication Efficient
> > > >     Decentralized Machine Learning\". But ICLR does not allow us to
> > > >     change the title during the rebuttal period.
> > > > 16. *Theorem 1: please point out to the definition of from Assumption 2
> > > >     in Theorem 1*
> > > >     **Reply:** We have made this addition.

---

> ### Author Response · Authors · 2023-11-21
> **Response Pt 5**
>
> 17.
>     **Reply**: We will update this shortly. We want to get the key
>     points from the review out, but will update this before the rebuttal
>     period has concluded.
>
> 18.
>     **Reply**: Unfortunately, to the best of our knowledge, we cannot
>     get anything if we do not have coercivity and Assumption 4 since
>     again this is required for analysis to hold. Furthermore, because we
>     do not preserve the average of the iterates so it is unclear how one
>     would do the analysis only using the assumptions in Koloskova et
>     al. 2021. We initially tried it this way and were unable to make any
>     progress.

---

> > ### Comment · Reviewer_zv9X · 2023-11-23
> > **Follow-up**
> >
> > Thank you for your rebuttal, I appreciate your attempt to address my concerns. Unfortunately, it's still difficult to be convinced of improvements in the paper. As the authors mention, in Table 1, in the worst case, there is no improvement. Sure, compared QSGD, their communication loss is smaller, is the factor significant enough? What is the maximum improvement and under which case it occurs? Compared to signSGD, there is no improvement. A bigger question is how realistic the considered setting is for the comparison?
> >
> > Moreover, the comparisons are also not always clear due to the difference in settings. signSGD and QSGD apply to general nonconvex setting (by just assuming smoothness) but in this paper there is additional assumptions (of course this paper focuses on a regularized problem whereas the previous ones focus on an unconstrained case).
> >
> > For this reason, it is also not convincing for the authors to say "whether or not $\ell_1$ regularization promotes sparsity is out of the scope" since this is one of the main ingredients that the authors say they use for improving communication complexity, this is the reason that the paper is focusing on regluarized problem and not unconstrained. Page 2 states for example "Our algorithm leverages model sparsification to reduce the model dimension by incorporating l1 regularization during the SGD update". App A.3 seems to assume a prior by assuming that $\ell_1$ norm will produce the required sparsity. This is not proving that $\ell_1$ regularization is doing what the authors claim it is doing, this looks more like assuming what they claim. This is what I understand in the paragraph "Bit Analysis Under Laplace Inputs", you can correct me if I am wrong. But in either case, it seems that the improvement is either quite small or there is no improvement compared to signSGD. I can see that experiments somewhat suggest some improvement, but they seem to be limited.
> >
> > > "Eq. (51) of our manuscript. We tackle this challenge by bounding the additional error terms using the result in Theorem 1 of the manuscript."
> >
> > Claims like this are difficult to verify. How do you bound these additional terms? Theorem 1 that uses 2 other lemmas, and it is complicated for a reader to parse the "main idea" from these and the information the authors provide. If the additional terms that do not appear in previous work are bounded by using the additional assumption (Assumption 4), this is not "tackling a challenge" but more of adding an assumption so that the challenge goes away. For example the rebuttal states "We cannot prove this lemma because we have the compression errors and hence why we introduce Assumption 4." If I am wrong and there are key ideas for tackling the new difficulties, these should be fleshed out.
> >
> > The last point is perhaps less important because as I said before, the theoretical contribution is difficult to quantify due to the lack of improvements in the bounds and assumptions being different from the previous work. As such, unfortunately, I cannot suggest acceptance at this stage. Going forward, I really recommend the authors to make the writing much clearer. This includes explaining every time a new assumption made, "why is this assumption reasonable" by giving proper references and real-world example settings (referring to Assumption 4 or coercivity). The same for results, they should make a clear case why the result is interesting and is better in some way than previous results. If the main improvement is illustrated empirically, then this should be clearly stated that the aim of the theory is to show there is some guarantees, but the real improvements are shown empirically. Then, the paper will attract more empirically-minded reviewers who can judge better the rigor of the experimental comparisons.

---

### Official Review · Reviewer_aRgR · 2023-11-07

**Soundness:** 3 good
**Presentation:** 2 fair
**Contribution:** 2 fair
**Rating:** 3
**Confidence:** 4

**Summary:**

The paper considers the non-convex decentralized learning problems with finite-sum, smooth, and non-convex loss functions with L1 regularization. The authors propose MALCOM-PSGD algorithm that strategically integrates gradient compression techniques with model sparsification. The proposed algorithm is guaranteed to converge at a sublinear rate with a diminishing stepsize. Numerical results are provided to show the advantages of the algorithm on saving of communication.

**Strengths:**

1. The proposed algorithm employs the residual compression via quantization and source coding methods to encode model sparsity, which can efficiently reduce the communication cost.
2. They provide a comprehensive convergence analysis of the MALCOM-PSGD algorithm, and its performance is substantiated by suitable experimental evidence.

**Weaknesses:**

1. The idea for model sparsification with L1 regularized training loss function and presented non-smooth problem are not surprising in distributed learning, which has been widely studied in the literature.
2. The communication complexity is not provided; the authors is suggested to compare the communication complexity with related works.
3. The theoretical results fail to demonstrate the existence of a linear-speedup case in decentralized training.
4. Assumption 4 is directly imposed on the sequence generated by the algorithm, which is not well justified and appears to be stringent.
5. The proof techniques used in the paper are standard in decentralized learning and source code methods is not original as well.
6. The algorithm design should be further clarified. Additionally, there is an abuse of notations, e.g. the constant $L$.

**Questions:**

refer to Weaknesses part.

**Details Of Ethics Concerns:**

N.A.

---

> ### Author Response · Authors · 2023-11-21
> **Response Pt 1**
>
> we thank the reviewer for the comments.
>
> 1.  *The idea for model sparsification with L1 regularized training loss
>     function and presented non-smooth problem are not surprising in
>     distributed learning, which has been widely studied in the
>     literature.*
>
>     **Reply**: We agree that model sparsification and proximal methods
>     have been extensively studied in distributed learning. However, it
>     is important to note that most of the existing research primarily
>     examines the influence of model sparsification for robustness or
>     dimension reduction without incorporating bit analysis of
>     compression.
>
>     Our research, in contrast, takes a distinctive approach of
>     *integrating* model sparsification with quantization and compression
>     techniques in local model communication. This approach significantly
>     enhances the communication efficiency in decentralized learning,
>     achieving a non-trivial $75$% gain compared to the state-of-the-art
>     method published in this conference \[Ref 3\]. Note this reference
>     also has established a "non-surprising\" result in the convergence
>     analysis. However, it is non-trivial to show that the same
>     convergence rate can be maintained with a non-smooth proximal
>     optimization step. Note that this also means their proof can not be
>     directly used in our case. To summarize, we demonstrate that 1)
>     model sparsification and quantization/compression interactively
>     impact the convergence in decentralized learning, and 2) model
>     sparsification can significantly improve the communication rate
>     during model compression when the low-entropy nature of model
>     updates has been taken into account.
>
>     In the revised Introduction, we have highlighted the above
>     contributions and insights, particularly emphasizing the coupled
>     effect of model sparsification and compressed model communication,
>     done with a vector quantization scheme specifically designed for
>     inputs with a fast-decaying probability for large values and sparse
>     support. This quantization scheme itself significantly helps even
>     other learning methods, such as \[Ref 3\], although it clearly
>     performs better when model sparsification is incorporated.
>
> 2.  *The communication complexity is not provided; the authors is
>     suggested to compare the communication complexity with related
>     works.*
>
>     **Reply**: In Section 5, we have expanded the discussion on the
>     communication complexity of the compression scheme under a concrete
>     example. Equation (18) shows that the communication bits required
>     for each local communication converge to
>     $\mathcal{O}(de^{-1/\Gamma})$ at the rate of
>     $\mathcal{O}(-d\log t)$. Here, $d$ represents the size of the model
>     parameters, $\Gamma$ represents the quantization levels, and $t$
>     denotes the training iteration. The decrease in communication bits
>     over iterations results from the decreasing of the model residual
>     values, which is an effect of model sparsification and the
>     convergence of the algorithm, making the residual smaller and
>     smaller.
>
>     In addition, we have included in Appendix A.3 a detailed comparison
>     of our compression scheme with other methods. The following table
>     shows the comparison in terms of the *worst-case* communication rate
>     with existing compressed schemes \[Ref 1\] and \[Ref 2\].
> | Compression method | \# of bits per local communication                                 |
> |--------------------|--------------------------------------------------------------------|
> | Our scheme         | Eq. (17) $< d(H(f^t)+	\text{const})\leq d(\log \Gamma+\text{const})$ |
> | QSGD [Ref 1]       | $d (\log \Gamma +\text{const})$                                    |
> | signSGD [Ref 2]    | $d$                                                                |
>
> We see from the table that the communication cost of our method
>     attains less bits compared to the method in \[Ref 1\]. This aligns
>     with our numerical findings in Section 6, which shows that our
>     approach consistently outperforms \[Ref 1\] in reducing
>     communication costs, exemplified by Fig. 1(c). When comparing our
>     method with signSGD from \[Ref 2\], with a small $\Gamma$, our
>     method exhibits comparable communication costs. Nevertheless, the
>     method in \[Ref 2\] is limited to one-bit quantization only, which
>     incurs substantial quantization errors and subsequently slower
>     training convergence. As demonstrated in Section 6, our method not
>     only maintains similar communication costs but also significantly
>     enhances training performance compared to \[Ref 2\].

---

> > ### Author Response · Authors · 2023-11-21
> > **Response Pt 2**
> >
> > 3.  *The theoretical results fail to demonstrate the existence of a
> >     linear-speedup case in decentralized training.*
> >
> >     **Reply**: We interpret the "linear-speedup\" as referring to the
> >     convergence rate improvement relative to the number of devices $n$.
> >     To the best of our understanding, the most relevant result in this
> >     context is presented in \[Ref 3, Theorem 4.1\]. This work shows that
> >     for decentralized SGD with a non-convex, unregularized, and smooth
> >     loss, the convergence rate, characterized by the average norm square
> >     of the gradients, is given by
> >     $\sum_{k=0}^{t} ||\nabla f(\bar {\bf x}^{(t)})||_2^2/(t+1)=\mathcal{O}(1/\sqrt{nt})$,
> >     where $t$ represents the number of training iterations.
> >
> >     First and foremost, we note that the goal of this work is to enhance
> >     communication efficiency rather than pursuing a linear speedup in
> >     terms of $n$. Our method has been demonstrated to achieve a
> >     substantial gain in reducing the communication delay for each
> >     communication round. Considering that our method achieves the same
> >     convergence rate as \[Ref 3\] in terms of the number of training
> >     iterations $t$. Therefore, our method achieves a much smaller
> >     *total* communication time. Second, the communication cost per
> >     training iteration grows with the increase in the number of devices
> >     $n$, because more devices lead to an increase in more model
> >     transmissions. It is crucial to note that even if one has a
> >     decreasing convergence rate with respect to $n$, this does not
> >     necessarily imply an overall reduction in the total communication
> >     cost.
> >
> >     Regarding the speedup of the convergence with respect to $n$, in
> >     contrast to the findings in \[Ref 3, Theorem 4.1\], our Theorem 2
> >     shows that the weighted average of the objective value
> >     $\bar{\mathcal{F}}_t$, *inclusive* of the non-smooth
> >     $\ell_1$-regularization term, converges towards the minimum
> >     objective at the rate of
> >     $\mathcal{O}\left(\frac{\ln t}{\sqrt{t}}(1+1/\sqrt{n})\right)$. This
> >     indicates a diminishing speedup with respect to $n$. To elucidate
> >     this discrepancy, we highlight the following key differences between
> >     our work and \[Ref 3\]:
> >
> >     -   Different setups: \[Ref 3\] optimizes a *smooth* loss by
> >         decentralization SGD, while our work optimizes a *non-smooth
> >         $\ell_1$-regularized* loss via decentralized *proximal* SGD. The
> >         non-smooth proximal step enlarges the error bound, thereby
> >         slowing down the training convergence in worst-case scenarios.
> >
> >     -   Different definitions of convergence: \[Ref 3\] focuses on the
> >         convergence rate towards a *stationary point*. In contrast, our
> >         analysis pertains to the convergence of the objective function
> >         to its minimum value, which is a more stringent criterion.
> >         Technically, they are not comparable with the different
> >         definitions.
> >
> >     Besides, our definition of convergence is in line with the analysis
> >     in \[Ref 4, Theorem 4(e)\]. This reference indicates that
> >     decentralized proximal SGD, when applied to a *convex* loss and
> >     executed with error-free model communication, converges at a rate of
> >     $\mathcal{O}(\frac{1}{n\sqrt{t}})$. In contrast, our analysis yields
> >     a significantly slower speedup rate with respect to $n$, due to the
> >     absence of convexity in the loss function.
> >
> >     On page 7 of the revised manuscript, we have discussed the speedup
> >     effect of $n$, as
> >
> >     "The convergence order
> >     $\mathcal{O}\left((1+1/\sqrt{n}) \ln t/\sqrt{t}\right)$ stated in
> >     Theorem 2 indicates a diminishing speedup with an increase in the
> >     number of devices $n$. This rate is notably slower than the rate of
> >     $\mathcal{O}(\frac{1}{n\sqrt{t}})$ achieved by decentralized PSGD
> >     with a convex loss and error-free communication, as demonstrated in
> >     \[Ref 4\]. The slower rate is attributed to the non-convex nature of
> >     the loss function in (1). Moreover, our analysis, focusing on the
> >     convergence of the non-smooth objective value $\mathcal{F}$ to its
> >     minimum, does not exhibit a linear speedup in terms of $n$, unlike
> >     the analysis in \[Ref 3\] which examines the convergence of the
> >     gradient vector of a smooth loss.\"

---

> > > ### Author Response · Authors · 2023-11-21
> > > **Response Pt 3**
> > >
> > > 4.  *Assumption 4 is directly imposed on the sequence generated by the
> > >     algorithm, which is not well justified and appears to be stringent.*
> > >
> > >     **Reply**: Please note that Assumption 4 has been revised to align
> > >     more cohesively with the other assumptions. This assumption requires
> > >     that the norm of the sequence generated by our algorithm,
> > >     $||{\bf X}^{(t)}||_F$, should be finite. Given in Assumption 3(ii)
> > >     that the objective function $\mathcal{F}$ is coercive[^1], the
> > >     boundedness of $||{\bf X}^{(t)}||_F$ is inherently assured if the
> > >     objective value in each iteration remains finite, i.e.,
> > >     $\mathcal{F}({\bf X}^{(t)})<\infty$. It is important to highlight
> > >     that this is a practical and reasonable assumption, as the values of
> > >     the loss function in a valid machine learning solver are expected to
> > >     be finite.
> > >
> > >     On Page 7, we have added more explanation for Assumption 4, as
> > >
> > >     " Assumption 3(ii) is consistent with \[Ref 4\] and under this
> > >     condition Assumption 4 requires that the loss values produced in
> > >     training iterations of our algorithm are finite. This assumption can
> > >     be reliably met with practical machine learning solvers. We note
> > >     that Assumption 4 is not required for \[Ref 3\] and \[Ref 4\], but
> > >     it is critical for our analysis since we need it to bound the
> > >     optimality gap. \[Ref 3\] is able to avoid this assumption by
> > >     assuming a smooth objective function while \[Ref 4\] avoids it by
> > >     assuming loss-less communication. \"
> > >
> > > 5.  *The proof techniques used in the paper are standard in
> > >     decentralized learning and source code methods is not original as
> > >     well.*
> > >
> > >     **Reply**: We address your concern from the following two aspects.
> > >
> > >     **Challenges in convergence analysis**. We highlight two major
> > >     challenges encountered during our convergence analysis. First, the
> > >     non-smooth nature of our objective function $\mathcal{F}$, coupled
> > >     with the consensus policy in Line 8 of Algorithm 1, presents a
> > >     unique challenge. Unlike the existing work such as \[Ref 3\], our
> > >     consensus recursion varies over time, with
> > >     $\overline {\bf X}^{(t)}\neq \overline {\bf X}^{(t+1)},\forall t$.
> > >     This variation prohibits us from directly bounding
> > >     $\mathbb{E}[\mathcal{F}(\overline {\bf X}^{(t+1)})]$ around
> > >     $\mathbb{E}[\mathcal{F}(\overline {\bf X}^{(t)})]$ using smoothness
> > >     properties. Consequently, our analysis has to focus on individual
> > >     objectives $\mathbb{E}[\mathcal{F}({\bf X}^{(t)})]$, resulting in a
> > >     more involved convergence proof.
> > >
> > >     Second, since we study non-convex decentralized optimization with
> > >     accumulative quantization errors over iterations, it becomes
> > >     infeasible to bound the objective gap between the local models
> > >     ${\bf X}^{(t)}$ and their average $\overline {\bf X}^{(t)}$ using
> > >     standard inequalities from convexity, as done in \[Ref 4, Eq.
> > >     (74)\]. Our approach bounds the consensus error represented by
> > >     $||{\bf X}^{(t)}-\overline {\bf X}^{(t)}||_F$ to effectively address
> > >     this challenge.
> > >
> > >     In summary, as the above challenges prevent directly adopting the
> > >     existing results, we believe that our analysis is noteworthy. It
> > >     offers novel techniques for analyzing decentralized optimization in
> > >     scenarios involving certain non-convex, non-smooth losses and random
> > >     communication errors. With these insights, we hope you will agree
> > >     with the meaningful contributions our research brings to the field.
> > >
> > >     **Our contributions to source encoding**. While the residual
> > >     encoding component in our algorithm draws directly from the work in
> > >     \[Ref 6\], our research introduces two innovative developments to
> > >     the source coding scheme. The method in \[Ref 6\] is designed for
> > >     encoding discrete inputs, with its analysis grounded in the
> > >     knowledge of the probability distribution of such discrete values.
> > >     In contrast, our work deals with model residuals in decentralized
> > >     learning, which are high-dimensional continuous random vectors
> > >     evolving throughout the training iterations. A key contribution of
> > >     our study is the theoretical quantification of the shrinkage rate of
> > >     these model residuals, followed by a tractable expression for the
> > >     communication bits. For additional details, please refer to the
> > >     response to Comment 2 above.
> > >
> > >     Furthermore, our findings also confirm that model sparsification
> > >     significantly enhances the compression rate in the studied source
> > >     encoding process. We believe this observation is also of notable
> > >     importance in the field of decentralized learning. Note another
> > >     paper \[Ref 1\] that simply adapts a known compression scheme, Elias
> > >     coding, in distributed learning. We believe our work provides more
> > >     substantial novel contributions by using a state-of-the-art
> > >     compression scheme that just appeared on ArXiv.

---

> > > > ### Author Response · Authors · 2023-11-21
> > > > **Response Pt 4**
> > > >
> > > > 6.  *The algorithm design should be further clarified. Additionally,
> > > >     there is an abuse of notations, e.g. the constant $L$.*
> > > >
> > > >     **Reply**: We have updated Section 3 to highlight the distinct
> > > >     elements of our algorithm compared to existing related approaches.
> > > >     Additionally, we have refined the notations: we use $L$ to represent
> > > >     the Lipschitz constant and $\Gamma$ to denote the number of
> > > >     quantization levels.
> > > >
> > > > References:
> > > >
> > > > [Ref 1]: Dan Alistarh, Demjan Grubic, Jerry Li, Ryota Tomioka, and Milan
> > > > Vojnovic. QSGD: Communication-efficient SGD via gradient quantization
> > > > and encoding. In *Advances in Neural Information Processing Systems*,
> > > > volume 30, pp. 1709--1720, 2017.
> > > >
> > > > [Ref 2]: Jeremy Bernstein, Yu-Xiang Wang, Kamyar Azizzadenesheli, and Anima
> > > > Anandkumar. SignSGD: Compressed optimisation for non-convex problems. In
> > > > *Proceedings of the 35th International Conference on Machine Learning*,
> > > > volume 80, pp. 560--569, July 2018.
> > > >
> > > > [Ref 3]: Anastasia Koloskova, Tao Lin, Sebastian U Stich, and Martin Jaggi.
> > > > Decentralized deep learning with arbitrary communication compression. In
> > > > *International Conference on Learning Representations*, volume 130, pp.
> > > > 2350--2358, 2021.
> > > >
> > > > [Ref 4]: Jinshan Zeng and Wotao Yin. On nonconvex decentralized gradient
> > > > descent. arXiv preprint arXiv:1608.05766, 2018.
> > > >
> > > > [Ref 5]: Sebastian U Stich. Local SGD converges fast and communicates little.
> > > > In *International Conference on Learning Representations*, 2019.
> > > >
> > > > [Ref 6]: Leah Woldemariam, Hang Liu, and Anna Scaglione. Low-complexity vector
> > > > source coding for discrete long sequences with unknown distributions.
> > > > arXiv preprint arXiv:2309.05633, 2023.

---

> > > > > ### Comment · Reviewer_aRgR · 2023-11-23
> > > > > **Response to the Rebuttal**
> > > > >
> > > > > The reviewer thank the authors's effort in the reply which have partially addressed the reviewer's concerns. The reviewer would thus maintain the score.

---

### Author Response · Authors · 2023-11-21
**Comment to all reviewers.**

We thank the reviewers for the time and effort spent in reading our paper and also for the constructive comments. We have addressed all the comments in the following response and made corresponding revisions in the manuscript. All the changes in the manuscript are colored in blue. We hope that the updated manuscript has well addressed all the comments.

Here are the **main changes in the manuscript**:
1.  Main context:
    1.  Enhanced the Introduction with additional discussions on our contributions, emphasizing the combined impact of model sparsification and gradient compression.
    2.  Enhanced the summary of our contributions with detailed explanations of the bit rate improvement of the compression scheme
    3. Expanded Section 3 with more detailed explanations of our algorithm.
    4. Provided more explanations for Assumption 4 and Theorem 2 in Section 4.
    5. Added communication complexity analysis for the compression scheme in Section 5.
    6. Introduced signSGD as a new baseline for numerical comparison in Section 6.
    7. Added simulation results to demonstrate the impact of model sparsification.
    8. Added more numerical results with various network setups.
2.  Appendices:
    1.  Added an extended analysis of communication bits and comparisons with other methods in Appendix A.3.
    2. Added more simulations to highlight the impact of different choices on the learning rate.

**The individual replies below** are our point-by-point responses to the review comments. The references used in the response are appended at the end of each post.

---

### Meta-Review · Area_Chair_c8ub · 2023-12-06

**Metareview:**

The review team raised several issues on this paper, including the following.

1. A linear speedup is not achieved by the proposed algorithm.

2. The algorithm presented seems to merge several existing methods like residual compression, communication and encoding, consensus aggregation, and proximal optimization. No new technique is developed in the components of the algorithm.

3. Additionally, the theoretical analysis seems to following to a standard steps with not much new technical development.

4. Assumption 4 is a little strong. Although the coercivity of $F$ helps, we still need to assume $\{F(x_t)\}$ is bounded, which is also a strong assumption that needs to be justified.

**Justification For Why Not Higher Score:**

Please see the meta review

**Justification For Why Not Lower Score:**

N/A

---

### Decision · Program_Chairs · 2024-01-16

Reject